# Beyond Counting Linear Regions of Neural Networks, Simple Linear Regions Dominate!

## Abstract

Functions represented by a neural network with the widely-used ReLU activation are piecewise linear functions over linear regions (polytopes). Figuring out the properties of such polytopes is of fundamental importance for the development of neural networks. So far, either theoretical or empirical studies on polytopes stay at the level of counting their number. Despite successes in explaining the power of depth and so on, counting the number of polytopes puts all polytopes on an equal booting, which is essentially an incomplete characterization of polytopes. Beyond counting, here we study the shapes of polytopes via the number of simplices obtained by triangulations of polytopes. First, we demonstrate the properties of the number of simplices in triangulations of polytopes, and compute the upper and lower bounds of the maximum number of simplices that a network can generate. Next, by computing and analyzing the histogram of simplices across polytopes, we find that a ReLU network has surprisingly uniform and simple polytopes, although these polytopes theoretically can be rather diverse and complicated. This finding is a novel implicit bias that concretely reveals what kind of simple functions a network learns and sheds light on why deep learning does not overfit. Lastly, we establish a theorem to illustrate why polytopes produced by a deep network are simple and uniform. The core idea of the proof is counter-intuitive: adding depth probably does not create a more complicated polytope. We hope our work can inspire more research into investigating polytopes of a ReLU neural network, thereby upgrading the knowledge of neural networks to a new level.

## 1 Introduction

It was shown in a thread of studies Chu et al. (2018); Balestriero & Baraniuk (2020); Hanin & Rolnick (2019b); Schonsheck et al. (2019) that a neural network with the piecewise linear activation is to partition the input space into many convex regions, mathematically referred to as polytopes, and each polytope is associated with a linear function (hereafter, we use convex regions, linear regions, and polytopes interchangeably). Hence, a neural network is essentially a piecewise linear function over the input domain. Based on this adorable result, the core idea of a variety of important theoretical advances and empirical findings is to turn the investigation of neural networks into the investigation of polytopes. By addressing basic questions such as how common operations affect the formation of polytopes (Zhang & Wu, 2020), how the network topology affects the number of polytopes (Cohen et al., 2016; Poole et al., 2016; Xiong et al., 2020), and so on, the understanding to expressivity of the networks is greatly deepened. To demonstrate the utility of the study on polytopes, we present two representative examples as follows:

The first representative example is the explanation to the power of depth. In the era of deep learning, many studies (Mohri et al., 2018; Bianchini & Scarselli, 2014; Telgarsky, 2015; Arora et al., 2016) attempted to explain why a deep network can perform superbly over a shallow one. One explanation to this question is on the superior representation power of deep networks, *i.e.*, a deep network can express a more complicated function but a shallow one with a similar size cannot (Cohen et al., 2016; Poole et al., 2016; Xiong et al., 2020). Their basic idea is to characterize the complexity of the function expressed by a neural network, thereby demonstrating that increasing depth can greatly maximize such a complexity measure compared to increasing width. Currently, the number of linear regions is one of the most popular complexity measures because it respects the functional structure

of the widely-used ReLU networks. Pascanu et al. (2013) firstly proposed to use the number of linear regions as the complexity measure. By directly applying Zaslavsky's Theorem (Zaslavsky, 1997), Pascanu et al. (2013) obtained a lower bound $\left(\prod_{l=0}^{L-1}\left\lfloor\frac{n_l}{n_0}\right\rfloor\right)\sum_{i=0}^{n_0}\binom{n_L}{i}$ for the maximum number of linear regions of a fully-connected ReLU network with $n_0$ inputs and $L$ hidden layers of widths $n_1, n_2, \cdots, n_L$. Since this work, deriving the lower and upper bounds of the maximum number of linear regions becomes a hot topic (Montufar et al., 2014; Telgarsky, 2015; Montúfar, 2017; Serra et al., 2018; Croce et al., 2019; Hu & Zhang, 2018; Xiong et al., 2020). All these bounds suggest the expressive ability of depth.

The second interesting example is the finding of the high-capacity-low-reality phenomenon (Hu et al., 2021; Hanin & Rolnick, 2019b), that the theoretical tight upper bound for the number of polytopes is much larger than what is actually learned by a network, *i.e.*, deep ReLU networks have surprisingly few polytopes both at initialization and throughout the training. Specifically, Hanin & Rolnick (2019b) proved that the expected number of linear regions in a ReLU network is bounded by a function of the number of total neurons and the input dimension. This counter-intuitive phenomenon can also be regarded as an implicit bias, which to some extent suggests why a deep network does not overfit. Although theoretically a lot of linear regions can be generated to learn a task, a deep network tends to find a simple function for a given task that is with few polytopes.

Despite figuring out the properties of polytopes of a neural network is of fundamental importance for the understanding of neural networks, the current studies on the polytopes have an important limit. So far, either theoretical or empirical studies only stay at the level of counting the number of polytopes, which blocks us from gaining other valuable findings. As we know, in a feed-forward network of $L$ hidden layers, each polytope is encompassed by a group of hyperplanes, as shown in Figure 1(a), and each hyperplane is associated with a neuron. The details of how polytopes are formed in a ReLU network can be referred to in Appendix A. Hence, any polytope

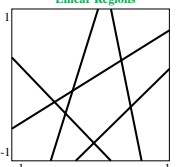 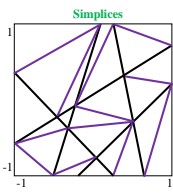

Figure 1: The number of simplices a polytope contains can reveal the shape information of a polytope, with which one can dig out valuable information of a neural network.

is created by at most $\sum_{i=1}^{L} n_i$ and at least $n_0 + 1$ hyperplanes, which is a large range. Face numbers of polytopes can vary a lot. Unfortunately, the existing "counting" studies did not accommodate the differences among polytopes. Therefore, it is highly necessary to move a step forward, *i.e.*, know what each polytope is, thereby capturing a more complete picture of a neural network. To realize so, as a first attempt, we seamlessly divide each polytope into simplices in a triangulation of the polytope, and we describe the shape of polytopes by the minimum number of simplices to partition it, as Figure 1 shows. For example in $\mathbb{R}^2$, if a polytope comprises three simplices, it is a pentagon.

In this manuscript, 1) to demonstrate the utility of the total number of simplices (#simplices) relative to the total number of polytopes (#polytopes), we characterize the basic proprieties and estimate the lower and upper bounds of the maximum #simplices for ReLU networks. The key to bound estimation is to estimate the total sum of the number of faces for all polytopes. 2) We observe that polytopes formed by ReLU networks are surprisingly uniform and simple. Here, the uniformity and simplicity mean that although theoretically quite diverse and complicated polytopes can be derived, simple polytopes dominate, *i.e.*, deep networks tend to find a function with a uniform and simple polytope pattern instead of a complicated polytope pattern. This is another high-capacity-low-reality phenomenon and an implicit simplicity bias of a neural network, implying how fruitful it is to go beyond counting. Previously, Hanin & Rolnick (2019b) showed that deep ReLU networks have few polytopes. Our report is that polytopes are not only few but also simple and uniform. Compared to (Hanin & Rolnick, 2019b), our observation more convincingly illustrates why deep networks do not overfit. Showing the number of polytopes is few is insufficient to claim that a network learns a simple solution because a network can have a small number of very complicated polytopes. 3) We establish a theorem that bounds the average face numbers of polytopes of a network to a small number under some mild assumption, thereby illustrating why polytopes produced by a deep network are simple and uniform.

To summarize, our contributions are threefold. 1) We point out the limitation of counting #polytopes. To solve it, we propose to use the #simplices to investigate the shape of polytopes. Investigating polytopes of a network can lead to a more complete characterization of ReLU networks and upgrade the knowledge of ReLU networks to a new level. 2) We empirically find that a ReLU network

has surprisingly uniform and simple polytopes. Such an interesting finding is a new implicit bias from the perspective of linear regions, which can shed light on why deep networks tend not to overfit. 3) To substantiate our empirical finding, we mathematically derive a tight upper bound for the average face number of polytopes, which not only offers a theoretical guarantee but also deepens our understanding of how a ReLU network partitions the space.

## 2 RELATED WORK

**Studies on polytopes of a neural network.** Besides the aforementioned works (Pascanu et al., 2013; Xiong et al., 2020; Montufar et al., 2014; Hu & Zhang, 2018) that count the number of polytopes, there are increasingly many studies on polytopes of neural networks. Chu et al. (2018); Hanin & Rolnick (2019b); Balestriero & Baraniuk (2020) showed that polytopes created by a network are convex. Zhang & Wu (2020) studied how different optimization techniques influence the local properties of polytopes, such as the inspheres, the directions of the corresponding hyperplanes, and the relevance of the surrounding regions. Hu et al. (2020) studied the network using an arbitrary activation function. They first used a piecewise linear function to approximate the given activation function. Then, they monitored the change of #polytopes to probe if the network overfits. Park et al. (2021) proposed the so-called neural activation coding that maximizes the number of linear regions to enhance the model's performance. Our work moves one step forward from counting the number of polytopes to considering the variability in shapes of polytopes, in hope to delineate a more complete picture of neural networks.

**Implicit bias of deep learning.** A network used in practice is highly over-parameterized compared to the number of training samples. A natural question is often asked: why do deep networks not overfit? To address this question, extensive studies have proposed that a network is implicitly regularized to learn a simple (not more expressive than necessary) solution. Implicit regularization is also referred to as an implicit bias. Gradient descent algorithms are widely believed to play an essential role in capacity control even when it is not specified in the loss function (Gunasekar et al., 2018; Soudry et al., 2018; Arora et al., 2019a; Sekhari et al., 2021; Lyu et al., 2021). Du et al. (2018); Woodworth et al. (2020) showed that the optimization trajectory of neural networks stays close to the initialization with the help of neural tangent kernel theory. Both theoretical derivation Tu et al. (2016); Li et al. (2020) and empirical findings Jing et al. (2020); Huh et al. (2021) suggested that a deep network tends to find a low-rank solution. To explain why deep networks first learn "simple patterns", a line of works Arora et al. (2019b); Cao et al. (2019); Yang & Salman (2019); Choraria et al. (2022) have analyzed the bias of a deep network towards lower frequencies. In contrast, our investigation identifies a new implicit bias from the perspective of linear regions. Different from most implicit biases highlighting a certain property of a network, our implicit bias concretely and straightforwardly reveals what kind of simple functions a network tends to learn.

## 3 PRELIMINARIES AND BASIC PROPERTIES

### 3.1 PRELIMINARIES

Throughout this paper, we always assume that the input space of an NN is a $d$-dimensional hypercube $C(d, B) := [-B, B]^d = \{\mathbf{x} = (x_1, x_2, \ldots, x_d) \in \mathbb{R}^d : -B \leq x_i \leq B\}$ for some large enough constant $B$.

Furthermore, we need the following definition for linear regions (polytopes).

**Definition 1** (Linear regions (polytopes) Hanin & Rolnick (2019a); Xiong et al. (2020)). *Suppose that $\mathcal{N}$ is a ReLU NN with $L$ hidden layers and input dimension $d$. An activation pattern of $\mathcal{N}$ is a function $\mathcal{P}$ from the set of neurons to the set $\{1, -1\}$, i.e., for each neuron $z$ in $\mathcal{N}$, we have $\mathcal{P}(z) \in \{1, -1\}$. Let $\theta$ be a fixed set of parameters in $\mathcal{N}$, and $\mathcal{P}$ be an activation pattern. Then the region corresponding to $\mathcal{P}$ and $\theta$ is*

$$\mathcal{R}(\mathcal{P}; \theta) := \{X \in C(d, B) : z(X; \theta) \cdot \mathcal{P}(z) > 0, \quad \forall z \text{ a neuron in } \mathcal{N}\},$$

*where $z(X; \theta)$ is the pre-activation of a neuron $z$. A linear region (polytope) of $\mathcal{N}$ at $\theta$ is a nonempty set $\mathcal{R}(\mathcal{P}, \theta) \neq \emptyset$ for some activation pattern $\mathcal{P}$. Let $R_{\mathcal{N}, \theta}$ be the number of linear regions of $\mathcal{N}$ at $\theta$, i.e., $R_{\mathcal{N}, \theta} := \#\{\mathcal{R}(\mathcal{P}; \theta) : \mathcal{R}(\mathcal{P}; \theta) \neq \emptyset \text{ for some activation pattern } \mathcal{P}\}$. Moreover, let $R_{\mathcal{N}} := \max_\theta R_{\mathcal{N}, \theta}$ denote the maximum number of linear regions of $\mathcal{N}$ when $\theta$ ranges over $\mathbb{R}^{\#weights + \#bias}$.*

In the following, Preliminary 1 shows that the polytopes created by a ReLU network are convex, which is the most important preliminary knowledge used in this manuscript. Since each polytope of a ReLU network is convex, as Figure 1 shows, one can further divide each polytope into simplices in a triangulation of polytopes to make it a simplicial complex (Preliminary 2), where a simplex is a fundamental unit. The number of simplices contained by a polytope can reflect the shape and complexity of the polytope. Then, Preliminary 3 introduces how to compute the vertices of polytopes. The detailed explanation of Preliminaries 1 and 3 can be seen in Appendix A.

**Preliminary 1** (Polytopes of a neural network are convex). *A neural network with ReLU activation partitions the input space into many polytopes (linear regions), such that the function represented by this neural network becomes linear when restricted in each polytope (linear region). Each polytope corresponds to a collection of activation states of all neurons, and each polytope is convex (Chu et al., 2018). In this paper, we mainly focus on $(n_0 - 1)$-dim faces of a $n_0$-dim polytope. **For convenience, we just simply use the terminology face to represent an $(n_0 - 1)$-dim facet of an $n_0$-dim polytope.***

**Preliminary 2** (Simplex and simplicial complex). *A **simplex** is just a generalization of the notion of triangles or tetrahedrons to any dimensions. More precisely, a D-simplex S is a D-dimensional convex hull provided by convex combinations of $D + 1$ affinely independent vectors $\{\mathbf{v}_i\}_{i=0}^D \subset \mathbb{R}^D$. In other words, $S = \left\{ \sum_{i=0}^D \xi_i \mathbf{v}_i \mid \xi_i \geq 0, \sum_{i=0}^D \xi_i = 1 \right\}$. The convex hull of any subset of $\{\mathbf{v}_i\}_{i=0}^D$ is called a face of S. A **simplicial complex** $\mathcal{S} = \bigcup_\alpha S_\alpha$ is composed of a set of simplices $\{S_\alpha\}$ satisfying: 1) every face of a simplex from $\mathcal{S}$ is also in $\mathcal{S}$; 2) the non-empty intersection of any two simplices $S_1, S_2 \in \mathcal{S}$ is a face of both $S_1$ and $S_2$. A **triangulation of a polytope** P is a partition of P into simplices such that: The union of all simplices equals P, and the intersection of any two simplices is a common face or empty.*

**Preliminary 3** (Computation of a polytope). *Given a ReLU network of L hidden layers, a collection of activation states of all neurons leads to a group of inequalities. Mathematically, a polytope with dimension $n_0$ is defined as $\{\mathbf{x} \in \mathbb{R}^{n_0} \mid \mathbf{a}_k \mathbf{x}^\top + b_k \leq 0, k \in [K]\}$, where $K = \sum_{i=1}^{L-1} n_i$ and $n_i$ is the number of neurons in the i-th layer. Given a group of inequalities, the vertices of the polytope can be computed based on the vertex enumeration algorithm (Avis & Fukuda, 1992).*

### 3.2 CHARACTERIZING PROPERTIES OF #SIMPLICES

• **Reflecting the shape variability of polytopes**: We plot the histogram of the number of simplices a polytope contains for a randomly initialized network of the structure 3-20-10-1 in Figure 2. As can be seen, the most complicated polytope comprises 40 simplices, while the simplest one only has 3 simplices, suggesting that polytopes formed by the network vary significantly. Naturally, these polytopes should not be regarded as equally complex. But when counting #polytopes, polytopes varying a lot are automatically treated equally. In contrast, since how many simplices a linear region comprises indicates how complex a linear region is, #simplices can characterize the differences of polytopes in shapes and further reveal more information about what kind of functions a ReLU network learns.

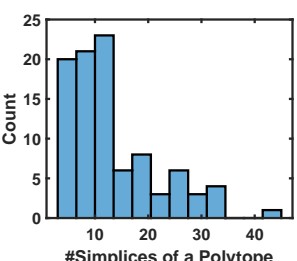

Figure 2: The histogram of #simplices with respect to different polytopes.

• **Estimating the bound of the maximum #simplices.** Here, we estimate the upper and lower bounds of the maximum #simplices of a ReLU network. Such results help us grasp the properties of #simplices and serve as a solid base for future theoretical or empirical applications of #simplices, such as understanding the generalization behavior of a network, estimating the computation time when using #simplices in experiments, *etc.* The detailed proofs are put into Appendix B.

**Theorem 1** (Upper Bound). *Let $\mathcal{N}$ be a multi-layer fully-connected ReLU NN with d input features and L hidden layers with n hidden neurons in each layer. Then the number of simplices in triangulations of all polytopes generated by $\mathcal{N}$ is at most*

$$\frac{2n^{dL}}{(d-1)!(d!)^{L-1}} + \mathcal{O}(n^{dL-1}). \tag{1}$$

*In particular, if $L = 1$, we derive the following upper bound for the maximum number of simplices*

$$\#simplices \leq 2n \sum_{i=0}^{d-1} \binom{n-1}{i} + 2d \sum_{i=0}^{d-1} \binom{n}{i}.$$

**Theorem 2** (Lower Bound). *Let $\mathcal{N}$ be a multi-layer fully-connected ReLU NN with $d$ input features and $L$ hidden layers with $n$ neurons in each layer. Then the maximum number of simplices in triangulations of polytopes generated by $\mathcal{N}$ is at least*

$$\frac{n^{dL}}{d^{d(L-1)}d!} + \mathcal{O}(n^{dL-1}).$$

*Furthermore, if $L = 1$, we derive the following tighter lower bound for the maximum number of simplices*

$$\#simplices \geq \frac{2n}{d+1} \sum_{i=0}^{d-1} \binom{n-1}{i}.$$

The basic idea to derive the above upper bound depends on the following observation: for each $(d-1)$-dim face of a $d$-dim polytope, it can only be a face for one unique simplex in a triangulation of this polytope, thus the total number of simplices in triangulations of polytopes must be smaller than or equal to the total number of $(d-1)$-dim faces in all polytopes. Therefore, we just need to derive the upper bound for the total number of $(d-1)$-dim faces in all polytopes generated by a neural network $\mathcal{N}$, which can be done by induction on the number of layers of $\mathcal{N}$. For the lower bound, we use the fact that each $d$-simplex with dimension $d$ has $d + 1$ faces, thus the number of simplices should be at least the total number of $(d-1)$-dim faces in all polytopes divided by $d + 1$.

We empirically validate our bounds in Table 1 with 7 structures for a comprehensive evaluation. For a network structure X-$Y_1$-$\cdots$-$Y_H$-1, X represents the dimension of the input, and $Y_h$ is the number of hidden neurons in the $h$-th hidden layer. For a given MLP architecture, we initialize all the parameters based on the Xavier uniform initialization. Because all network structures we validate have a limited number of neurons, we can compute polytopes and their simplices by enumerating all collective activation states of neurons, which ensures that all polytopes are identifiable. For each structure, we repeat initialization ten times to report the maximum #simplices. As shown in Table 1, the derived upper bound is compatible with the numerical results of several network structures, which verifies the correctness of our results.

Table 1: Numerically verify the correctness of the derived upper and lower bounds for the maximum #simplices.

|  | 3-4-1 | 3-5-1 | 3-6-1 | 3-7-1 | 3-8-1 | 3-9-1 | 3-10-1 |
|---|---|---|---|---|---|---|---|
| Upper Bounds by Theorem 1 | 122 | 206 | 324 | 482 | 686 | 942 | 1256 |
| Estimation by Enumeration Method | 119 | 183 | 317 | 446 | 663 | 893 | 1140 |
| Lower Bounds by Theorem 2 | 14 | 27 | 48 | 77 | 116 | 166 | 230 |

## 4 DEEP ReLU NETWORKS HAVE SIMPLE AND UNIFORM LINEAR REGIONS

Previously, deep neural networks were both theoretically and empirically identified to have surprisingly fewer linear regions than their maximum capacity (Hanin & Rolnick, 2019b). By analyzing the number of simplices a polytope contains, we observe that linear regions formed by ReLU networks are surprisingly simple and uniform. Although theoretically quite diverse linear regions can be derived, simple linear regions dominate, which is another high-capacity-low-reality phenomenon (Hu et al., 2021) and a new implicit bias, which may explain why a deep learning model tends not to overfit. Combining the finding in (Hanin & Rolnick, 2019b), we can upgrade the conclusion to that deep ReLU networks have surprisingly *few*, *simple*, and *uniform* linear regions. Compared to other implicit biases emphasizing properties of a network such as capacity, rank, and frequency, we concretely show what kinds of simple functions a ReLU network learns. We validate our finding comprehensively at different initialization methods, network depths, sizes of the outer bounding box, biases, the bottleneck, network architecture, and input dimensions. Furthermore, we showcase that during the training, although the number of linear regions increases, linear regions keep their uniformity and simplicity. To ensure the preciseness of the discovery and be limited by the prohibitive computational cost of deriving simplices, our experiments are primarily on low-dimensional inputs.

## 4.1 INITIALIZATION

We validate four popular initialization methods: Xavier uniform, Xavier normal[1], Kaiming, orthogonal initialization (He et al., 2015). For each initialization method, we use two different network architectures (3-40-20-1, 3-80-40-1). The bias values are set to 0.01 for all neurons. A total of 8,000 points are uniformly sampled from $[-1, 1]^3$ to compute the polytope. At the same time, we check the activation states of all neurons to avoid counting some polytopes more than once. Each run is repeated five times. Figure 3 shows the histogram of the #simplices each polytope has. Hereafter, if no special specification, the x-axis of all figures denotes the number of simplices a polytope has, and the y-axis denotes the count of polytopes with a certain number of simplices. Without loss of generality, suppose that in an experiment, the maximum #simplices a polytope has is $\Omega$, we deem a polytope with no more than $\Omega/3$ as simple; otherwise, it is complex. The spotlight is that for all initialization methods and network structures, simple polytopes significantly dominate over complicated polytopes. We calculate that simple polytopes take account for at least 57% and at most 76% of the total. In addition, among different initialization methods, the Xavier normal method tends to generate more uniform polytopes on four architectures. The achieved polytope is far simpler than the theoretically most complicated polytope. In Appendices D-I, via systematic experiments, we find that the simple linear regions still take the majority at different network depths, sizes of the outer bounding box, biases, the bottleneck, network architecture, and input dimensions.

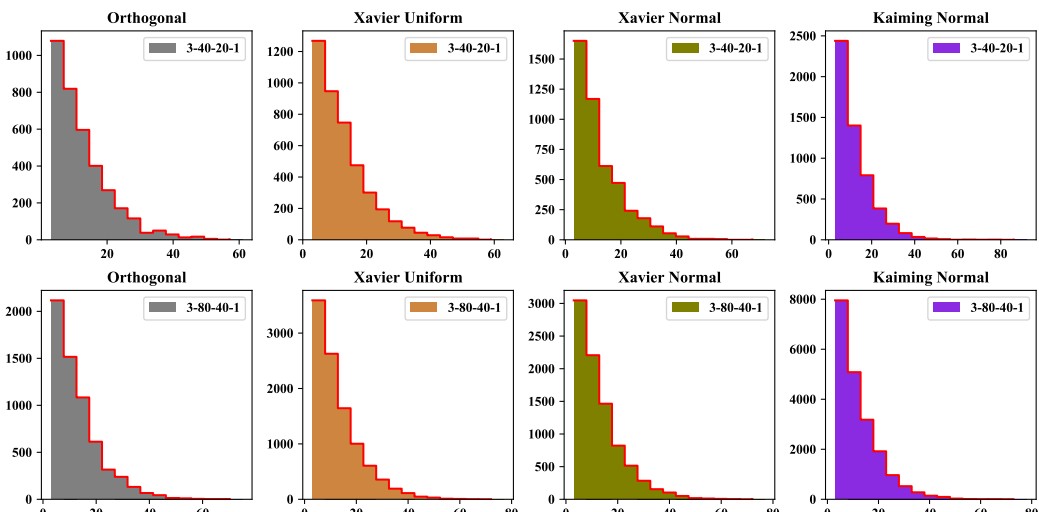

Figure 3: Deep ReLU networks have surprisingly uniform linear regions at different initialization methods.

## 4.2 TRAINING

Earlier, we show that at the initialization stage, deep networks exhibit simple and uniform linear regions. Next, it is natural to ask *will the uniformity and simplicity of linear regions be broken during training*? We answer this question by training a fully-connected network using ReLU activation function on a real-world problem and counting the simplices of polytopes. The task is to predict if a COVID-19 patient will be at high risk, given one's health status, living habits, and medical history. This prediction task has 388,878 raw samples, and each has 5 medical features including 'HIPERTENSION','CARDIOVASCULAR', 'OBESITY', 'RENAL CHRONIC', 'TOBACCO'. The labels are 'at risk' or 'no'. The detailed descriptions of data and this task can be referred to in Kaggle[2]. The data are preprocessed as follows: The discrete value is assigned to different attributes. If a patient has that pre-existing disease or habit, 1 will be assigned; otherwise, 0 will be assigned. Then, the data are randomly split into training and testing sets with a ratio of 0.8:0.2.

We implement a network of 5-20-20-1. The optimizer is Adam with a learning rate of 0.1. The network is initialized by Xavier uniform. The loss function is the binary cross-entropy function. The epoch number is 400 to guarantee convergence. A total of 8,000 points are uniformly sampled

---

[1]https://pytorch.org/docs/stable/nn.init.html

[2]https://www.kaggle.com/code/meirnizri/covid-19-risk-prediction

from $[-1, 1]^3$ to compute the polytope. The outer bounding box is $[-5, 5]^3$ to ensure as many polytopes as possible are counted. Figure 4 shows that as the training goes on, the number of linear regions drops compared to the random initialization. Furthermore, throughout the training, most polytopes are simple.

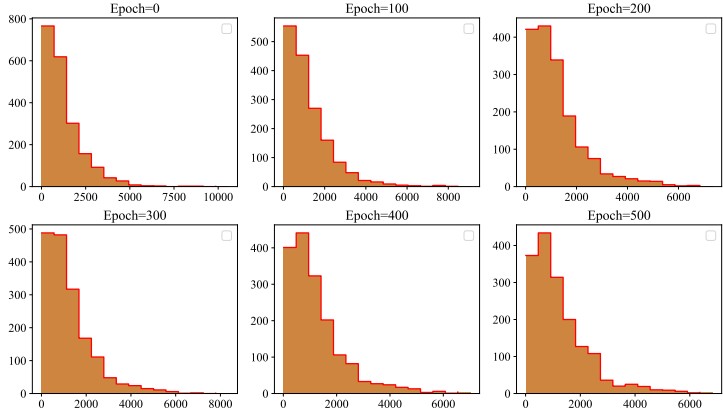

Figure 4: The results show that throughout the training, most polytopes are simple.

We also train networks on MNIST, following the same procedure in (Hanin & Rolnick, 2019b). Here, we can not compute the polytopes in $28 \times 28$ dimensional space because the vertex enumeration algorithm suffers the curse of dimensionality. Therefore, we visualize the polytopes in the cross-section plane. We initialize a network of size 784-7-7-6-10 with Kaiming normalization. The batch size is 128. The network is trained with Adam with a learning rate of 0.001. The total epoch number is set to 35, which ensures the sufficient training of the network.

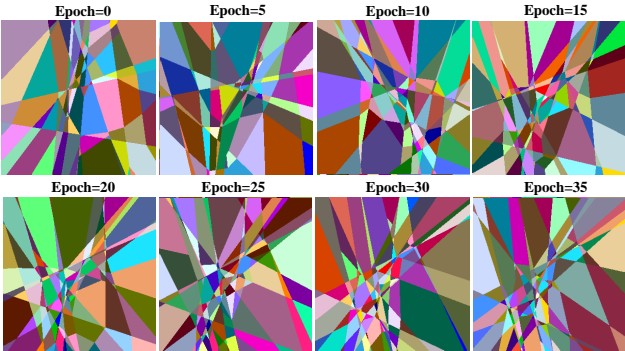

Figure 5: A cross-sectional visualization to the polytopes learned by a network at different epochs.

Figure 5 shows the cross-section of the function learned by a network at different epochs. The cross-section is a plane that pass through two randomly-selected images from MNIST. Figure 5 shows that as the training goes on, the number of polytopes increases. The spotlight is that almost all the polytopes are triangles or quadrilaterals. Although these polytopes are from a cross-section other than the whole landscape, one can indirectly sense the uniformity of these polytopes.

## 5 MATHEMATICAL INTERPRETATION

Here, we mathematically explain why simple polytopes dominate in ReLU networks.

**Geometric heuristics.** We attribute the uniformity and simplicity of polytopes to the locality created by the ReLU activation function. Since a ReLU network divides the space into many local polytopes, to yield a complicated polytope from a local polytope, two or more hyperplanes associated with neurons in the later layers should intersect within the given local polytope, which is hard because the area of polytopes is typically small. As such, the complexity of polytopes probably does not increase as the network goes deeper. Our heuristics are also supported by (Hanin & Rolnick, 2019b)'s proof, where a deep ReLU network was proved to have few polytopes because hyperplanes do not cross in a local polytope. Without crossing, complicated polytopes will not emerge either. Furthermore,

we believe the average face number is an intrinsic quantity of a ReLU network majorly due to the property of space partition and the network's hierarchical structure. Such an intrinsic quantity is essentially only weakly dependent on the width and depth of a network.

Now, we formalize our heuristic into a tight bound of the average face numbers of polytopes.

**Theorem 3** (One-hidden-layer NNs). *Let $\mathcal{N}$ be a one-hidden-layer fully-connected ReLU NN with $d$ inputs and $n$ hidden neurons, where $d$ is a fixed positive integer. Suppose that $n$ hyperplanes generated by $n$ hidden neurons are in general position. Let $C(d, B) := [-B, B]^d$ be the input space of $\mathcal{N}$. Furthermore, assume that $n$ and $B$ are large enough, then the average number of faces in linear regions of $\mathcal{N}$ is at most $2d + 1$.*

*Proof.* By Theorem 1, we obtain that the number of simplices in triangulations of polytopes generated by $\mathcal{N}$ is at most

$$\#\text{simplices} \leq 2n \sum_{i=0}^{d-1} \binom{n-1}{i} + 2d \sum_{i=0}^{d-1} \binom{n}{i}.$$

On the other hand, since the $n$ hidden neurons are in general position and $B$ is large enough, we obtain that the total number of polytopes (i.e., linear regions) produced by $\mathcal{N}$ is $\sum_{i=0}^{d} \binom{n}{i}$. Therefore, the average number of faces in linear regions of $\mathcal{N}$ is at most

$$\frac{2n \sum_{i=0}^{d-1} \binom{n-1}{i} + 2d \sum_{i=0}^{d-1} \binom{n}{i}}{\sum_{i=0}^{d} \binom{n}{i}} \leq \frac{2n \sum_{i=0}^{d-1} \binom{n-1}{i} + 2d \sum_{i=0}^{d-1} \binom{n}{i}}{\sum_{i=0}^{d-1} \binom{n}{i+1}}.$$

For each $0 \leq i \leq d - 1$, we have

$$\frac{2n \cdot \binom{n-1}{i} + 2d\binom{n}{i}}{\binom{n}{i+1}} \leq 2(i+1) + \frac{2d(i+1)}{n-i} = 2(i+1)\left(1 + \frac{d}{n-i}\right)$$

$$\leq 2d\left(1 + \frac{d}{n-d+1}\right).$$

Therefore, when $n$ is large enough, we have the average number of faces in linear regions of $\mathcal{N}$ is at most $2d + \mathcal{O}(\frac{1}{n}) \leq 2d + 1$. $\qquad\square$

**Theorem 4** (Multi-layer NNs). *Let $\mathcal{N}$ be an $L$-layer fully-connected ReLU NN with $d$ inputs and $n_i$ hidden neurons in the $i$-th hidden layer where $d$ is a fixed positive integer. Suppose that for each linear region $S$ produced by the first $(i-1)$-th layers of $\mathcal{N}$, the $n_S$ hyperplanes in $S$ generated by hidden neurons in the $i$-th layer are always in general position. Let $C(d, B) := [-B, B]^d$ be the input space of $\mathcal{N}$. Furthermore, assume that each $n_S$ and $B$ are large enough, then the average number of faces in linear regions of $\mathcal{N}$ is at most $2d + 1$.*

*Proof.* Please see Appendix B.5. $\qquad\square$

**Remark 1.** One desirable property about this bound is that it is independent of the width and depth, which validates our heuristics. Considering that $2d + 1$ is a rather small bound, Theorems 3 and 4 can justify why simple polytopes dominate. If the dominating polytopes are complex polytopes, the average face number should surpass $2d + 1$ a lot. If simple polytopes only take up a small portion, the average face number will be larger than $2d + 1$, too. Although we assume that the network is wide in deriving the bound, based on our geometric heuristics, the average face number should also be small for narrow networks. We leave this question for future exploration.

## 6 DISCUSSION AND CONCLUSION

**Implication to spectral bias**. Previously, a plethora of studies observed that deep networks first learn patterns of low frequencies (Arora et al., 2019b; Cao et al., 2019; Yang & Salman, 2019; Choraria et al., 2022). This observation is referred to as the spectral bias. Our finding is highly relevant to the spectral bias. Combined with the observation in (Hanin & Rolnick, 2019b), since polytopes are few, simple, and uniform, the function learned by a ReLU network does not produce a lot of oscillations in all directions, which roughly corresponds to a low-frequency solution. Our opinion is that the function represented by a ReLU network is initialized to express a function without too many oscillations. When the stride by gradient descent is small, a network naturally tends to first learn patterns of low frequencies.

**Decoding the learning of neural networks**. Going from the linear regions to the simplices also contributes to a novel understanding of the training of a neural network. It has been clear that the learning of a neural network is the process of the function fitting with a piecewise linear function. The view of simplices connects with the finite element method (Bathe, 2007). Since a simplex is an elementary unit, we argue that the learning of a neural network is nothing but formulating finite elements towards solving the stationarity of the functional:

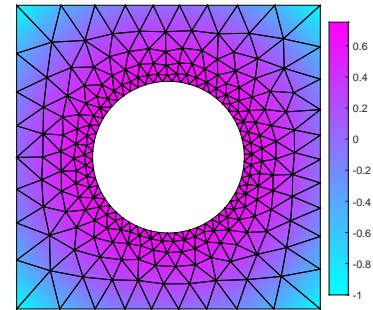

Figure 6: Decoding the learning of neural network as constructing finite element under the stationary functional.

$$\Pi = \int_\Omega R(u(\mathbf{x})) + \sum_k \delta(\mathbf{x} - \mathbf{x}_k)(u(\mathbf{x}) - \bar{u}(\mathbf{x}))d\mathbf{x}, \quad (2)$$

where $u(\mathbf{x}) = \sum_{i=1}^n \sum_{j=1}^d c_{ij} N_j(\mathbf{x})$, $\bar{u}(\mathbf{x})$ is the ground-truth function, $R(u(\mathbf{x}))$ denotes some regularization to $u(\mathbf{x})$, $N_j(\mathbf{x})$ is the shape function over a finite element (simplex) that is obtained through coordinate transform, and $c_{ij}$ are coefficient solved by $\frac{\partial \Pi}{\partial c_{ij}} = 0, i = 1, 2, \cdots, n, j = 1, 2, \cdots, d$. The viewpoint of constructing finite elements is highly related to moving mesh methods in the field of scientific computing (Di et al., 2005), which could inspire more discoveries in the future.

In this manuscript, we have advocated studying the properties of polytopes instead of just counting them, towards revealing other valuable properties of a neural network. To show such a direction is beneficial, we have characterized the desirable properties of #simplices and estimated its upper and lower bounds to pave the way for future applications. Then, we have presented that deep ReLU networks have surprisingly simple and uniform linear regions, which is an implicit bias for ReLU networks, and may explain why deep networks do not overfit. Lastly, we have mathematically established a small bound for the average number of faces in polytopes to explain our finding. In the future, more efforts should be put into investigating the polytopes of a network and designing advanced algorithms to calculate the polytopes.

## 7    REPRODUCIBILITY STATEMENT

To ensure that the experimental results and conclusions of our paper are totally reproducible, we make efforts as the following:

Theoretically, we state the full set of preliminaries, list all requisites of theorems, and include complete proofs of theorems in Sections 3 and 5; Appendices A and B.

Experimentally, we provide the code in the supplementary material. Our code contains all necessary comments and is highly readable to ensure that interested readers can reproduce it. In addition, we specify all the training and implementation details in Section 4.

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

# A  SIMPLICES, POLYTOPES, AND THEIR COMPUTATION

## A.1  NOTATIONS OF A NETWORK

For convenience and consistency, we inherit notations from (Chu et al., 2018). For a ReLU network that contains $L$ hidden layers, we write the $l$-th layer as $\mathcal{L}_l$. Specially, $\mathcal{L}_0$ is the input layer, $\mathcal{L}_{L+1}$ is the output layer, and the other layers $\mathcal{L}_l, l \in \{1, 2, \ldots, L\}$ are hidden layers. Hidden layers' neurons are called hidden neurons. Let $n_l$ represent the number of neurons in $\mathcal{L}_l$.

Given the $i$-th neuron of the $l$-th hidden layer, we denote by $\mathbf{b}_i^{(l)}$ its bias, by $\mathbf{a}_i^{(l)}$ its output, and by $\mathbf{z}_i^{(l)}$ the total weighted sum of its inputs plus the bias. For all the $n_l$ neurons in $\mathcal{L}_l$, we arrange all their biases into a vector $\mathbf{b}^{(l)} = \left[ \mathbf{b}_1^{(l)}, \ldots, \mathbf{b}_{n_l}^{(l)} \right]^\top$, their outputs into a vector $\mathbf{a}^{(l)} = \left[ \mathbf{a}_1^{(l)}, \ldots, \mathbf{a}_{n_l}^{(l)} \right]^\top$, and their inputs into a vector $\mathbf{z}^{(l)} = \left[ \mathbf{z}_1^{(l)}, \ldots, \mathbf{z}_{n_l}^{(l)} \right]^\top$. Neurons in two neighbor layers are linked by weighted edges. Denote by $W_{ij}^{(l)}$ the weight of the edge between the $i$-th neuron in $\mathcal{L}_{l+1}$ and the $j$-th neuron in $\mathcal{L}_l$. For $l \in \{1, \ldots, L\}$, we compute $\mathbf{z}^{(l+1)}$ by

$$\mathbf{z}^{(l+1)} = W^{(l)} \mathbf{a}^{(l)} + \mathbf{b}^{(l)}, \tag{3}$$

where $W^{(l)}$ is an $n_{l+1}$-by-$n_l$ matrix.

Let $\sigma : \mathbb{R} \to \mathbb{R}$ be the ReLU activation function. We have $\mathbf{a}_i^{(l)} = \sigma \left( \mathbf{z}_i^{(l)} \right)$ for all $l \in \{1, \ldots, L\}$. In an element-wise fashion, we write $\sigma \left( \mathbf{z}^{(l)} \right) = \left[ \sigma \left( \mathbf{z}_1^{(l)} \right), \ldots, \sigma \left( \mathbf{z}_{n_l}^{(l)} \right) \right]^\top$. Then, for $l \in \{1, \ldots, L\}$, we have

$$\mathbf{a}^{(l)} = \sigma \left( \mathbf{z}^{(l)} \right). \tag{4}$$

The input is $\mathbf{x} \in \mathcal{X}$, where $\mathcal{X} \in \mathbb{R}^d$, and $\mathbf{x}_i$ is the $i$-th dimension of $\mathbf{x}$. The input layer $\mathcal{L}_0$ contains $n_0 = d$ nodes, where $\mathbf{a}_i^{(0)} = \mathbf{x}_i, i \in \{1, \ldots, d\}$. The output of the network is $\mathbf{a}^{(L+1)} \in \mathcal{Y}$, where $\mathcal{Y} \subseteq \mathbb{R}^{n_{l+1}}$. The output layer $\mathcal{L}_{L+1}$ employs the softmax function: $\mathbf{a}^{(L+1)} = \mathrm{softmax} \left( \mathbf{z}^{(L+1)} \right)$.

## A.2  DERIVING POLYTOPES OF A NETWORK

For the $i$-th hidden neuron in $\mathcal{L}_l$, $\sigma \left( \mathbf{z}_i^{(l)} \right)$ is in the following form:

$$\sigma \left( \mathbf{z}_i^{(l)} \right) = \begin{cases} \mathbf{z}_i^{(l)}, & \text{if } \mathbf{z}_i^{(l)} \geq 0 \\ 0, & \text{if } \mathbf{z}_i^{(l)} < 0, \end{cases} \tag{5}$$

where $\sigma \left( \mathbf{z}_i^{(l)} \right)$ consists of two linear parts. Given a network, an instance $\mathbf{x} \in \mathcal{X}$ determines the value of $\mathbf{z}_i^{(l)}$, and further determines $\sigma \left( \mathbf{z}_i^{(l)} \right) = 0$ or $\sigma \left( \mathbf{z}_i^{(l)} \right) = \mathbf{z}_i^{(l)}$. According to which part $\sigma \left( \mathbf{z}_i^{(l)} \right)$ falls into, one can encode the activation status of each hidden neuron by two states, each of which uniquely corresponds to one part of $\sigma \left( \mathbf{z}_i^{(l)} \right)$. Denote by $\mathbf{c}_i^{(l)} \in \{1, 0\}$ the state of the $i$-th hidden neuron in $\mathcal{L}_l$, we have $\mathbf{z}_i^{(l)} \geq 0$ if and only if $\mathbf{c}_i^{(l)} = 1$ and $\mathbf{z}_i^{(l)} < 0$ if and only if $\mathbf{c}_i^{(l)} = 0$. The states of different hidden neurons usually differ from each other.

Let $\mathbf{c}^{(l)} = \left[ \mathbf{c}_1^{(l)}, \ldots, \mathbf{c}_{n_l}^{(l)} \right]$ be the states of all hidden neurons in $\mathcal{L}_l$ and $\mathbf{C} = \left[ \mathbf{c}^{(1)}, \ldots, \mathbf{c}^{(L)} \right]$ specify the collective states of all hidden neurons. $\mathbf{C}$ of a given fixed network is uniquely determined by the instance $\mathbf{x}$. We write the function that maps an instance $\mathbf{x} \in \mathcal{X}$ to a configuration $\mathrm{C} \in \{1, 0\}^N$ as $\mathrm{conf} : \mathcal{X} \to \{1, 0\}^N$, where $N = \sum_{i=0}^{L-1} n_l$. Then, we rewrite Eq. (4) as

$$\mathbf{a}^{(l)} = \sigma \left( \mathbf{z}^{(l)} \right) = \mathbf{c}^{(l)} \circ \mathbf{z}^{(l)}, \tag{6}$$

where $\mathbf{c}^{(l)} \circ \mathbf{z}^{(l)}$ is the Hadamard product between $\mathbf{c}^{(l)}$ and $\mathbf{z}^{(l)}$.

By plugging $\mathbf{a}^{(l)}$ into Eq. (3), we rewrite $\mathbf{z}^{(l+1)}$ as

$$\mathbf{z}^{(l+1)} = W^{(l)} \left( \mathbf{c}^{(l)} \circ \mathbf{z}^{(l)} \right) + \mathbf{b}^{(l)} = \tilde{W}^{(l)} \mathbf{z}^{(l)} + \tilde{\mathbf{b}}^{(l)}, \tag{7}$$

where $\tilde{\mathbf{b}}^{(l)} = \mathbf{b}^{(l)}$ , and $\tilde{W}^{(l)} = W^{(l)} \circ \mathbf{c}^{(l)}$ is the generalized Hadamard product, such that the entry at the $i$-th row and $j$-th column of $\tilde{W}^{(l)}$ is $\tilde{W}_{ij}^{(l)} = W_{ij}^{(l)} \mathbf{c}_j^{(l)}$.

By iteratively enforcing Eq. (7), we can write $\mathbf{z}^{(l+1)}, l \in \{1, \dots, L\}$ as

$$\mathbf{z}^{(l+1)} = \left( \prod_{k=1}^{l} \tilde{W}^{(k)} \right) \mathbf{z}^{(1)} + \sum_{h=1}^{l} \left( \prod_{k=h+1}^{l} \tilde{W}^{(k)} \right) \tilde{\mathbf{b}}^{(h)}. \tag{8}$$

Substituting $\mathbf{z}^{(1)} = W^{(0)} \mathbf{x} + \mathbf{b}^{(1)}$ into the above equation, we rewrite $\mathbf{z}^{(l+1)}, l \in \{1, \dots, L\}$ as

$$
\begin{aligned}
\mathbf{z}^{(l+1)} &= \left( \prod_{k=1}^{l} \tilde{W}^{(k)} \right) (W^{(0)} \mathbf{x} + \mathbf{b}^{(1)}) + \sum_{h=1}^{l} \left( \prod_{k=h+1}^{l} \tilde{W}^{(k)} \right) \tilde{\mathbf{b}}^{(h)} \\
&= \left( \prod_{k=0}^{l} \tilde{W}^{(k)} \right) \mathbf{x} + \mathbf{b}^{(1)} \left( \prod_{k=1}^{l} \tilde{W}^{(k)} \right) + \sum_{h=1}^{l} \left( \prod_{k=h+1}^{l} \tilde{W}^{(k)} \right) \tilde{\mathbf{b}}^{(h)} \\
&= \hat{W}^{(0:l)} \mathbf{x} + \hat{\mathbf{b}}^{(0:l)},
\end{aligned}
\tag{9}
$$

where $\hat{W}^{(0:l)}$ is the coefficient matrix of $\mathbf{x}$, and $\hat{\mathbf{b}}^{(0:l)}$ is the sum of the remaining bias terms. Naturally, $F(\mathbf{x})$ is

$$F(\mathbf{x}) = \mathrm{softmax} \left( \hat{W}^{(0:L)} \mathbf{x} + \hat{\mathbf{b}}^{(0:L)} \right). \tag{10}$$

For a fixed network and a fixed instance $\mathbf{x}$, $\hat{W}^{(0:l)}$ and $\hat{\mathbf{b}}^{(0:l)}$ are constant parameters uniquely determined by activation states of all neurons: $\mathbf{C} = \mathrm{conf}(\mathbf{x})$. Furthermore, according to Eq. (5), $\mathbf{C}$ leads to a group of inequalities:

$$(2\mathbf{c}^{(l+1)} - 1) \circ (\hat{W}^{(0:l)} \mathbf{x} + \hat{\mathbf{b}}^{(0:l)}) \geq 0, \; l = 0, \cdots, L - 1, \tag{11}$$

which encompasses a convex polytope according to the $H$-definition of polytopes (Chapter 16, Toth et al. (2017)). If $\mathrm{conf}(\mathbf{x}_1) = \mathrm{conf}(\mathbf{x}_2)$, $\mathbf{x}_1$ and $\mathbf{x}_2$ lie in the same polytope due to the uniqueness.

### A.3 Computing #Simplices and Polytopes with Vertex Enumeration Algorithm

---

**Algorithm 1** Calculate the #simplex of a polytope of a neural network

---

1: Identify a collective activation state of all neurons
2: Derive a group of inequalities whose hyperplanes encompass the targeted polytope
3: Vertex enumeration algorithm to derive the vertices of the polytope
4: Delaunay triangulation
5: Count #simplices

---

Next, what is essential is to numerically compute #simplices. Unfortunately, the exact computation relies on enumerating all collective states of neurons, which is only possible i) when the total number of neurons in a network is small or ii) when the input dimension is low. For the former, we can enumerate possible activation states of all neurons by repeating $2^{\sum_{i=0}^{L-1} n_i}$ times, while for the latter, we can uniformly sample the input space by $(\frac{1}{\epsilon})^d$ times, where $\epsilon$ is the needed sampling interval, to identify the actual collective states of all neurons. As highlighted earlier, the collective activation states $\mathbf{C}$ lead to a group of inequalities, corresponding to a polytope. We can solve all vertices of the polytope with the vertex enumeration algorithm [3] (Avis & Fukuda, 1992) to find all vertices of a polytope. The complexity of the vertex enumeration algorithm scales linearly with the number of inequalities. Then, Delaunay triangulation [Chapter 23, (Toth et al., 2017)] is executed for these vertices to see how many non-overlapping simplices these vertices can gain. Lastly, we count the #simplices. Sometimes, one may not need to compute all polytopes of a neural network. Instead, the polytopes related to data are sufficient. For example, in the training-free NAS experiment (Chen et al., 2021), the authors exhausted all training data and identify associated unique polytopes.

---

[3] https://pypi.org/project/pypoman/

# B ESTIMATION TO THE MAXIMUM #SIMPLICES

## B.1 PRELIMINARY

Let's recall some basic knowledge on hyperplane arrangements (Stanley et al., 2004). Let $V$ be a Euclidean space. A hyperplane in the Euclidean space $V \simeq \mathbb{R}^n$, is a subspace $H := \{X \in V : \alpha \cdot X = b\}$, where $\mathbf{0} \neq \alpha \in V, b \in \mathbb{R}$ and "$\cdot$" denotes the inner product. A *region* of an arrangement $\mathcal{A} = \{H_i \subset V : 1 \leq i \leq m\}$ is just a connected component in the complement set of the union of all hyperplanes in the arrangement $\mathcal{A}$. Let $r(\mathcal{A})$ be the number of regions for an arrangement $\mathcal{A}$. Also, a *simplex* in an $n$-dimensional Euclidean space is just a $n$-dimensional polytope that is the convex hull of $n + 1$ vertices. For example, a triangle is a simplex in $\mathbb{R}^2$, and a tetrahedron is a simplex in $\mathbb{R}^3$. A *triangulation* on some polytope is a division of the polytope into into simplices.

The following Zaslavsky's Theorem is very crucial in the estimation of the number of linear regions.

**Lemma 1** (Zaslavsky's Theorem (Zaslavsky, 1975; Stanley et al., 2004))**.** *Let $\mathcal{A}$ be an arrangement with $m$ hyperplanes in $\mathbf{R}^n$. Then, the number $r(\mathcal{A})$ of regions for the arrangement $\mathcal{A}$ satisfies*

$$r(\mathcal{A}) \leq \sum_{i=0}^{n} \binom{m}{i}. \tag{12}$$

*Furthermore, the above equality holds iff $\mathcal{A}$ is in general position Stanley (2004).*

## B.2 MAIN RESULTS - ONE LAYER ReLU NNs

Throughout this paper, we always assume that the input space of an NN is a $d$-dimensional hypercube $C(d, B) := \{\mathbf{x} = (x_1, x_2, \ldots, x_d) \in \mathbb{R}^d : -B \leq x_i \leq B\}$ for some large enough constant $B$. Note that for a one-layer fully-connected ReLU NN, the pre-activation of each hidden neuron is an affine linear function of input values. Based on the sign of the pre-activation, each hidden neuron produces a hyperplane that divides the input space into two linear regions. On the other hand, the $d$-dimensional hypercube $C(d, B)$ has $2d$ hyperplanes in its boundary.

**Theorem 5.** *Let $\mathcal{N}$ be a one-layer fully-connected ReLU NN with $d$ input features and $n$ hidden neurons. Then the number of simplices in triangulations of polytopes generated by $\mathcal{N}$ is at most*

$$2n \sum_{i=0}^{d-1} \binom{n-1}{i} + 2d \sum_{i=0}^{d-1} \binom{n}{i}.$$

*Proof.* Let $H_1, H_2, \ldots, H_n$ be the $n$ hyperplanes generated by $n$ hidden neurons and $H_{n+1}, H_{n+2}, \ldots, H_{n+2d}$ be the $2d$ hyperplanes in the boundary of $C(d, B)$. Then for each $1 \leq i \leq n$, the hyperplane $H_i$ may be intersected by other $n - 1$ hyperplanes in $H_1, H_2, \ldots, H_n$. This will produce at most $n - 1$ hyperplanes in $H_i$, thus by Theorem 1, it will divide $H_i$ into at most $\sum_{i=0}^{d-1} \binom{n-1}{i}$ pieces since $H_i$ is a $(d-1)$-dim hyperplane. Also, for each $1 \leq i \leq 2d$, the hyperplane $H_{n+i}$ may be intersected by $H_1, H_2, \ldots, H_n$. This will produce at most $n$ $(d-2)$-dim hyperplanes in $H_{n+i}$, thus by Theorem 1, it will divide $H_i$ into at most $\sum_{i=0}^{d-1} \binom{n}{i}$ pieces since $H_i$ is a $(d-1)$-dim hyperplane. Moreover, each piece could be a face of two linear regions, finally we will get at most

$$2n \sum_{i=0}^{d-1} \binom{n-1}{i} + 2d \sum_{i=0}^{d-1} \binom{n}{i}$$

faces for all the polytopes. On the other hand, each simplex in a triangulation of polytope can be corresponding to at least one face in the polytope, and each face in the polytope can be corresponding to exactly one simplex. Therefore, the total number of simplices must be smaller than or equal to the total number of faces in all polytopes. Thus we obtain that the number of simplices in triangulations of polytopes generated by $\mathcal{N}$ is also at most

$$2n \sum_{i=0}^{d-1} \binom{n-1}{i} + 2d \sum_{i=0}^{d-1} \binom{n}{i}.$$

$\square$

The following results gives a lower bound for the maximum number of simplices in a triangulation of for a one layer fully-connected ReLU NN.

**Theorem 6.** *Let $\mathcal{N}$ be a one-layer fully-connected ReLU NN with $d$ input features and $n$ hidden neurons. If $n$ corresponding hyperplanes are in general position and $C(d, B)$ is large enough, then the number of simplices in a triangulation of polytopes among all $n$ corresponding hyperplanes is at least*

$$\frac{2n}{d+1} \sum_{i=0}^{d-1} \binom{n-1}{i} = \frac{2n^d}{(d+1)(d-1)!} + \mathcal{O}(n^{d-1}).$$

*Proof.* Let $H_1, H_2, \ldots, H_n$ be $n$ hyperplanes generated by $n$ hidden neurons. Then for each $1 \leq i \leq n$, the hyperplane $H_i$ will be intersected by other $n-1$ hyperplanes in $H_1, H_2, \ldots, H_n$. This will produce exact $n-1$ hyperplanes in $H_i$ since $H_1, H_2, \ldots, H_n$ are in general position, thus by Theorem 1, it will divide $H_i$ into exact $\sum_{i=0}^{d-1} \binom{n-1}{i}$ pieces since $H_i$ is a $(d-1)$-dim hyperplane. When $C(d, B)$ is large enough, we can assume that every such a piece has a non-empty intersection with $C(d, B)$. Therefore, the total sum of number of $(d-1)$-faces of all linear regions (polytopes) will be at least $2n \sum_{i=0}^{d-1} \binom{n-1}{i}$ since every piece is counted twice. On the other hand, every $d$-dim simplex has $d+1$ distinct $(d-1)$-dim faces, thus every triangulation with $N$ simplices will contain $N(d+1)$ number $(d-1)$-dim faces. Therefore, if a triangulation of all linear regions (polytopes) of $\mathcal{N}$ contains $N$ simplices, then

$$N(d+1) \geq 2n \sum_{i=0}^{d-1} \binom{n-1}{i}$$

and thus

$$N \geq \frac{2n}{d+1} \sum_{i=0}^{d-1} \binom{n-1}{i}.$$

Finally, we derive that a triangulation of all linear regions (polytopes) of $\mathcal{N}$ contains at least $\frac{2n}{d+1} \sum_{i=0}^{d-1} \binom{n-1}{i} = \frac{2n^d}{(d+1)(d-1)!} + \mathcal{O}(n^{d-1})$ simplices. $\qquad\square$

### B.3 MAIN RESULTS - MULTI-LAYER RELU NNS

To study the multi-layer NNs, we need the following results from (Montúfar, 2017, Proposation 3).

**Lemma 2** (Montúfar (2017)). *Let $\mathcal{N}$ be a multi-layer fully-connected ReLU NN with $d$ input features and $L$ hidden layers with $n_1, n_2, \ldots, n_L$ hidden neurons. Then the number of polytopes of $\mathcal{N}$ is at most $\prod_{i=1}^{L} \sum_{j=0}^{m_i} \binom{n_i}{j}$, where $m_i = \min\{d, n_1, n_2, \ldots, n_i\}$.*

**Theorem 7.** *Let $\mathcal{N}$ be a multi-layer fully-connected ReLU NN with $d$ input features and $L$ hidden layers with $n$ hidden neurons in each layer. Then the number of simplices in triangulations of polytopes generated by $\mathcal{N}$ is at most*

$$\frac{2n^{dL}}{(d-1)!(d!)^{L-1}} + \mathcal{O}(n^{dL} - 1). \tag{13}$$

*Proof.* First, we prove by induction that the total number of faces generated by $\mathcal{N}$ is at most

$$\frac{2n^{dL}}{(d-1)!(d!)^{L-1}} + \mathcal{O}(n^{dL} - 1).$$

The case $L = 1$ is proved in Theorem 5. When $L \geq 2$, we assume that Eq. (13) holds for $L - 1$. Thus by Lemma 2, and the induction hypothesis, the network $\mathcal{N}'$ with the first $L - 1$ layers already has

$$\frac{n^{d(L-1)}}{(d!)^{L-1}} + \mathcal{O}(n^{d(L-1)-1})$$

linear regions and

$$\frac{2n^{d(L-1)}}{(d-1)!(d!)^{L-2}} + \mathcal{O}(n^{d(L-1)-1})$$

faces for all polytopes. Then when we add the $L$-th layer, for each polytope $R$ with $f_R$ faces in $\mathcal{N}'$, the $n$ neurons and the $f_R$ faces create at most $n + f_R$ hyperplanes in $R$, similar to Theorem 5 these creates

$$2n \sum_{i=0}^{d-1} \binom{n-1}{i} + f_R \sum_{i=0}^{d-1} \binom{n}{i}$$

faces for all the polytopes in $R$. Therefore, we obtain that the total number of faces is at most

$$2n \sum_{i=0}^{d-1} \binom{n-1}{i} \cdot \left( \frac{n^{d(L-1)}}{(d!)^{L-1}} + \mathcal{O}(n^{d(L-1)-1}) \right) + \sum_{i=0}^{d-1} \binom{n}{i} \sum_R f_R$$

$$= 2n \sum_{i=0}^{d-1} \binom{n-1}{i} \cdot \left( \frac{n^{d(L-1)}}{(d!)^{L-1}} + \mathcal{O}(n^{d(L-1)-1}) \right)$$

$$+ \sum_{i=0}^{d-1} \binom{n}{i} \cdot \left( \frac{2n^{d(L-1)}}{(d-1)!(d!)^{L-2}} + \mathcal{O}(n^{d(L-1)-1}) \right)$$

$$= \frac{2n^{dL}}{(d-1)!(d!)^{L-1}} + \mathcal{O}(n^{dL} - 1).$$

Therefore, the total number of simplices must be smaller than or equal to the total number of faces in all polytopes. Thus we obtain that the number of simplices in triangulations of polytopes generated by $\mathcal{N}$ is also at most

$$\frac{2n^{dL}}{(d-1)!(d!)^{L-1}} + \mathcal{O}(n^{dL} - 1).$$

$\square$

On the other hand, by the following lemma it is easy to derive the maximum number of simplices in triangulations of polytopes generated by multi-layer NNs.

**Lemma 3** (Montufar et al. (2014)). *Let $\mathcal{N}$ be a multi-layer fully-connected ReLU NN with $d$ input features and $L$ hidden layers with $n_l$ hidden neurons in the $l$-th layer. Then the maximum number of linear regions of $\mathcal{N}$ is at least $\prod_{l=1}^{L-1} \lfloor \frac{n_l}{d} \rfloor^d \sum_{j=0}^{d} \binom{n_L}{j}$.*

For the lower bounds, we have the following results.

**Theorem 8.** *Let $\mathcal{N}$ be a multi-layer fully-connected ReLU NN with $d$ input features and $L$ hidden layers with $n$ neurons in each layer. Then the maximum number of simplices in triangulations of polytopes generated by $\mathcal{N}$ is at least*

$$\frac{n^{dL}}{d^{d(L-1)}d!} + \mathcal{O}(n^{dL-1}).$$

*Proof.* By Lemma 3, the the maximum number of linear regions is lower bounded by $\left(\frac{n}{d}\right)^{d(L-1)} \sum_{i=0}^{d} \binom{n}{i} = \frac{n^{dL}}{d^{d(L-1)}d!} + \mathcal{O}(n^{dL-1})$. Also, the number of simplices should be larger than or equal to the number of linear regions. Thus we obtain the number of simplices in a triangulation of polytopes among all $n$ corresponding hyperplanes is at least $\frac{n^{dL}}{d^{d(L-1)}d!} + \mathcal{O}(n^{dL-1})$. $\square$

Now, we can derive Theorems 1 and 2 in the main text.

*Proof of Theorem 1.* Directly by Theorems 5 and 7. $\square$

*Proof of Theorem 2.* Directly by Theorems 6 and 8. $\square$

### B.4 COMPARISON OF DIFFERENT NETWORK ARCHITECTURES

The aim of this section is to compare the the maximum #simplices based on bounds obtained in Section B.2. Our conclusion is that deep NNs usually have more number of simplices than shallow NNs with the same number of parameters.

First let's fix some notations. For two functions $f(n)$ and $g(n)$, we write $f(n) = \Theta(g(n))$ if there exists some positive constants $c_1, c_2$ such that $c_1 g(n) \leq f(n) \leq c_2 g(n)$ for all sufficiently large $n$; $f(n) = \mathcal{O}(g(n))$ if there exists some positive constant $c > 0$ such that $f(n) \leq c g(n)$ for all sufficiently large $n$; and $f(n) = \Omega(g(n))$ if there exists some positive constant $c$ such that $f(n) \geq c g(n)$ for all sufficiently large $n$.

The number of parameters for the fully-connected ReLU NN $\mathcal{N}$ is easy to compute (Pascanu et al., 2013, Proposition 7).

**Lemma 4.** *Let $\mathcal{N}$ be a multi-layer fully-connected ReLU NN with $d$ input features and $L$ hidden layers with $n$ hidden neurons in each layer. Then the number of parameters in $\mathcal{N}$ is $\Theta(Ln^2)$.*

Let $S_{\mathcal{N}_1}$ be the maximum number of simplices in triangulations of polytopes generated by $\mathcal{N}$. Now we can derive the number of simplices per parameter for deep NNs and their shallow counterparts. The following result follows directly from Lemma 4, Theorem 1 and Theorem 2.

**Theorem 9.** *Let $\mathcal{N}_1$ be a multi-layer fully-connected ReLU NN with $d$ input features and $L$ hidden layers with $n$ hidden neurons in each layer, and $d = \mathcal{O}(1)$. Then $\mathcal{N}_1$ has $\Theta(Ln^2)$ parameters, and the ratio of $S_{\mathcal{N}_1}$ to the number of parameters of $\mathcal{N}_1$ is*

$$\frac{S_{\mathcal{N}_1}}{\# \text{ parameters of } \mathcal{N}_1} = \Omega\left(\frac{1}{L} \cdot \frac{n^{dL-2}}{d^{d(L-1)}d!}\right).$$

*For a one-layer fully-connected ReLU NN $\mathcal{N}_2$ with $d$ input features and $Ln^2$ hidden neurons, it has $\Theta(Ln^2)$ parameters, and the ratio for $\mathcal{N}_2$ is*

$$\frac{S_{\mathcal{N}_2}}{\# \text{ parameters of } \mathcal{N}_2} = \mathcal{O}\left(\frac{(Ln^2)^{d-1}}{(d-1)!}\right).$$

From Theorem 9 we obtain that $\frac{S_{\mathcal{N}_1}}{\# \text{ parameters of } \mathcal{N}_1}$ grows at least exponentially fast with the depth $L$ and polynomially fast with the width $n$. In contrast, $\frac{S_{\mathcal{N}_2}}{\# \text{ parameters of } \mathcal{N}_2}$ grows at most polynomially fast with the numbers $L$ and $n$.

Therefore, we have that $\frac{S_{\mathcal{N}_1}}{\# \text{ parameters of } \mathcal{N}_1}$ is far larger than $\frac{S_{\mathcal{N}_2}}{\# \text{ parameters of } \mathcal{N}_2}$ when $L$ and $n$ are sufficiently large. Thus we conclude that fully-connected ReLU NNs usually create much more number of simplices than one-layer fully-connected ReLU NNs with asymptotically the same number of input dimensions and parameters. This result suggests that fully-connected ReLU NNs usually have much more expressivity than one-layer fully-connected ReLU NNs.

### B.5 AVERAGE NUMBER OF FACES IN LINEAR REGIONS OF NEURAL NETWORKS

In this subsection, we provide the proofs of Theorem 4.

*Proof of Theorem 4.* Similar to the proof of Theorem 3, we obtain that for each linear region $S$ produced by the first $(i-1)$-th layers of $\mathcal{N}$, the average number of faces in linear regions of $S$ after adding the $i$-th layer is at most

$$\frac{2n_S \sum_{i=0}^{d-1} \binom{n_S-1}{i} + f \sum_{i=0}^{d-1} \binom{n_S}{i}}{\sum_{i=0}^{d} \binom{n_S}{i}} \leq \frac{2n_S \sum_{i=0}^{d-1} \binom{n_S-1}{i} + f \sum_{i=0}^{d-1} \binom{n_S}{i}}{\sum_{i=0}^{d-1} \binom{n_S}{i+1}}$$

where $f$ is the number of faces of $S$. For each $0 \leq i \leq d-1$, we have

$$\frac{2n_S \cdot \binom{n_S-1}{i} + f\binom{n_S}{i}}{\binom{n_S}{i+1}} \leq 2(i+1) + \frac{f(i+1)}{n_S - i} \leq 2d\left(1 + \frac{f}{2(n_S - d + 1)}\right).$$

When $n_S$ is large enough, $\frac{f}{2(n_S-d+1)}$ should tend to 0, thus the right hand side of the last inequality should be smaller than $2d + 1$. Therefore, the average number of faces in linear regions of $\mathcal{N}$ is also at most $2d + 1$. □

# C  OTHER PROPERTIES OF #SIMPLICES

• **One-to-one correspondence via modularization and network transform**: A simplex is an elementary unit. Proposition 1 shows that an arbitrary ReLU network can be transformed into a wide network whose depth is the input dimension and width is determined by the #simplices. Therefore, as long as the support of functions represented by two networks is filled by the same #simplices, two networks can be represented by the network of the same width and depth. As such, we can define the equivalent class of networks as those with the same #simplices to cover their polytopes. In contrast, #polytopes cannot be used to rigorously define the equivalent networks because the polytopes are not elementary units. When one is told that two networks have the same #polytopes, one is still not totally sure whether two networks are equally complex.

**Proposition 1** (#Simplices in network transform (Fan et al., 2020)). *Suppose that the representation of an arbitrary ReLU network is $h : [-B,\ B]^D \to \mathbb{R}$, and $M$ is the minimum #simplices to cover the polytopes to support $h$, for any $\delta > 0$, there exists a wide ReLU network $\mathbf{H}$ of width $\mathcal{O}\left[D(D+1)(2^D - 1)M\right]$ and depth $D$, satisfying that*

$$\mathfrak{m}\Big(\mathbf{x} \mid h(\mathbf{x}) \neq \mathbf{H}(\mathbf{x})\}\Big) < \delta \tag{14}$$

*where $\mathfrak{m}(\cdot)$ is the standard measure in $[-B,\ B]^D$.*

# D SUPPLEMENTARY EXPERIMENTS FOR DIFFERENT DEPTHS

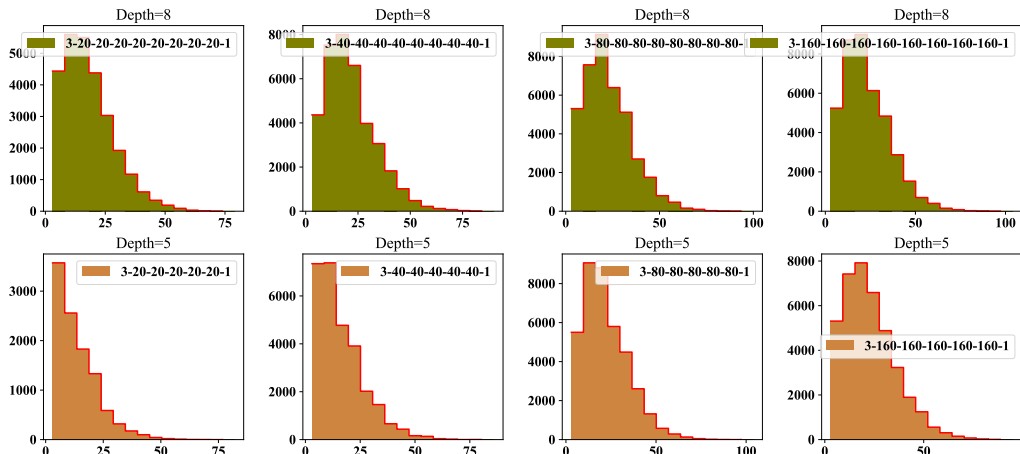

Figure 7: The uniformity and simplicity hold true for deep networks under Xavier uniform initialization.

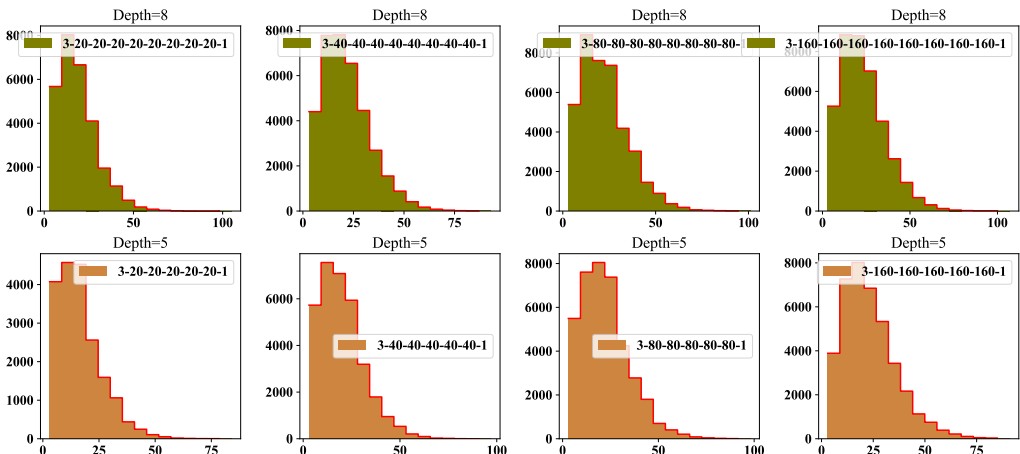

Figure 8: The uniformity and simplicity hold true for deep networks under Xavier normal initialization.

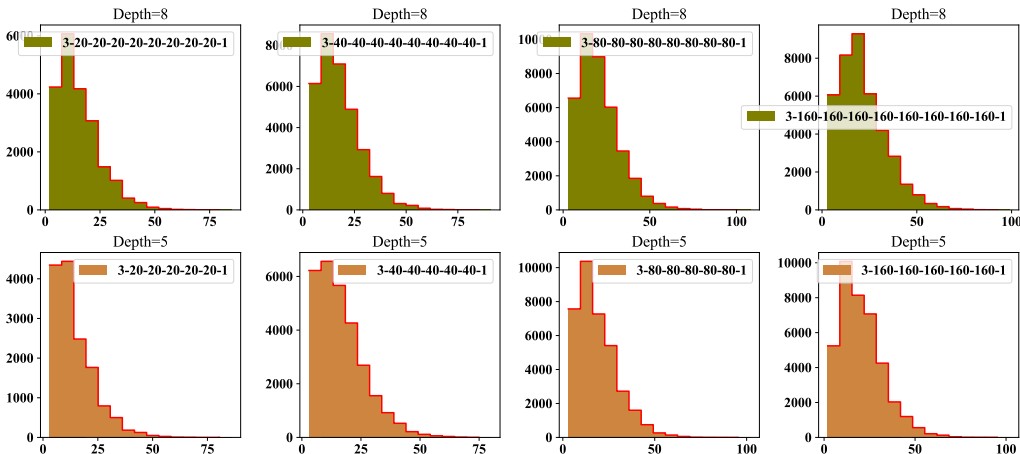

Figure 9: The uniformity and simplicity hold true for deep networks under Kaiming initialization.

Here, we evaluate if the uniformity and simplicity of polytopes still hold for deeper networks. This question is non-trivial, since a deeper network can theoretically generate more complicated polytopes. Will the depth break the uniformity? We choose four different widths (20, 40, 80, 160).

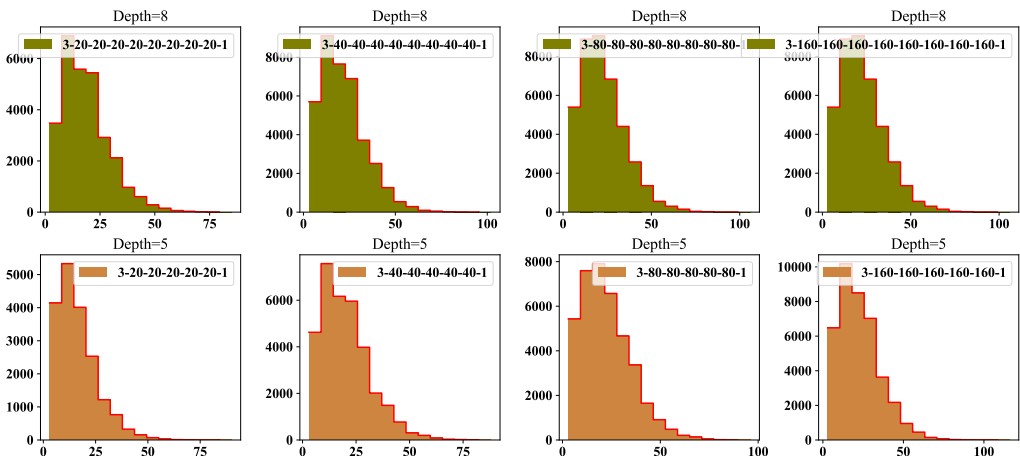

Figure 10: The uniformity and simplicity hold true for deep networks under orthogonal initialization.

For comprehensiveness, the network initialization methods are the Xavier uniform, Xavier normal, Kaiming, and orthogonal initialization. The depth is set to 5 and 8, respectively. The bias value is 0.01. Likewise, a total of 8,000 points are uniformly sampled from $[-1, 1]^3$ to compute the polytope. At the same time, we check the activation states of all neurons to avoid counting some polytopes more than once. Each run is repeated five times. The results under four different initialization are shown in Figures 7, 8, 9, and 10, from which we draw three highlights. First, we find that both going deep and going wide can increase the number of polytopes at different initializations. But the effect of going deep is much more significant than that of going wide. Second, when the network goes deep, although that the total number of polytopes increases, simple polytopes still dominate among all polytopes. Third, for different initialization methods and different depths, the dominating polytope is slightly different. For example, the dominating polytopes for the network 3-40-40-40-40-40-1 under Xavier normal initialization are those with 6∼10 simplices, while the dominating polytopes for the network 3-20-20-20-20-20-1 under Xavier uniform initialization are those with 1∼5 simplices.

# E    SUPPLEMENTARY EXPERIMENTS FOR DIFFERENT BIASES

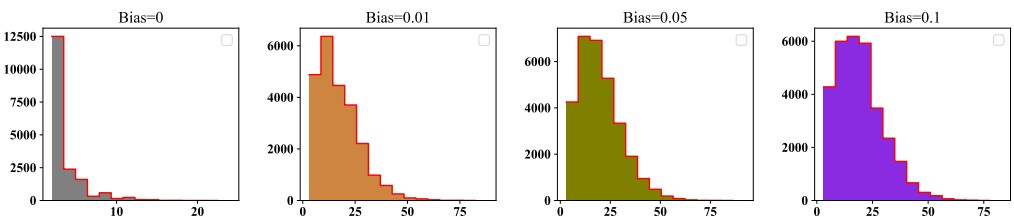

Figure 11: The simplicity and uniformity hold true for different bias values under Xavier initialization.

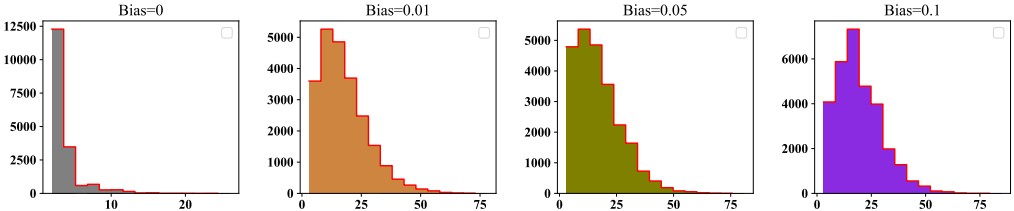

Figure 12: The simplicity and uniformity hold true for different bias values under Xavier normal initialization.

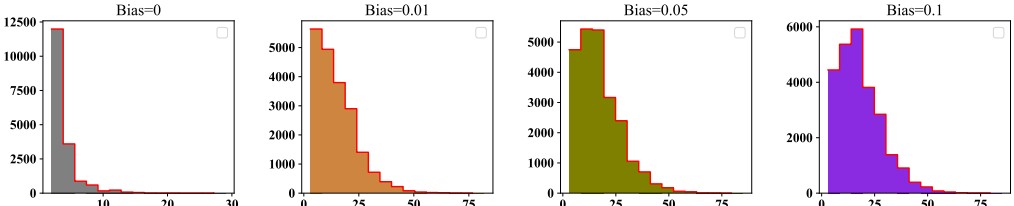

Figure 13: The simplicity and uniformity hold true for different bias values under Kaiming initialization.

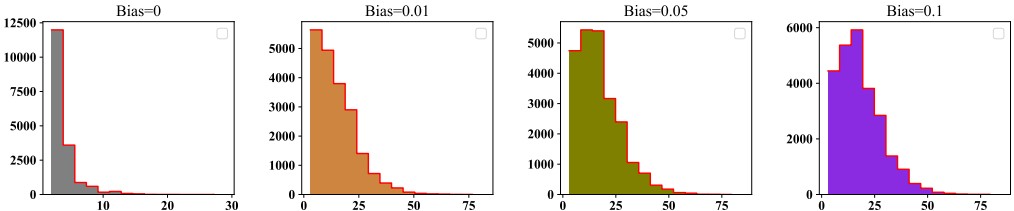

Figure 14: The simplicity and uniformity hold true for different bias values under orthogonal initialization.

Here, we are curious about how the bias value of neurons will affect the distribution of polytopes. To address this issue, we set the bias values to $0, 0.01, 0.05, 0.1$, respectively for the network 3-80-40-1. The outer bounding box is $[-1, 1]^3$. A total of 8,000 points are uniformly sampled from $[-1, 1]^3$ to compute the polytope. At the same time, we check the activation states of all neurons to avoid counting some polytopes more than once. Each run is repeated five times. The initialization methods are the Xavier uniform, Xavier normal, Kaiming, and orthogonal initialization. As shown in Figures 11, 12, 13, and 14 for different initialization methods, we observe that as the bias value increases, more polytopes are produced. However, the number of simple polytopes still takes up the majority. It is worthwhile mentioning that when the bias equals 0, the uniformity is crystal clear. The bias=0 is the extremal case, where all hyperplanes of the first layer intersect at the original point, and much fewer facets in polytopes are created.

# F SUPPLEMENTARY EXPERIMENTS FOR BOUNDING BOXES OF DIFFERENT SIZES

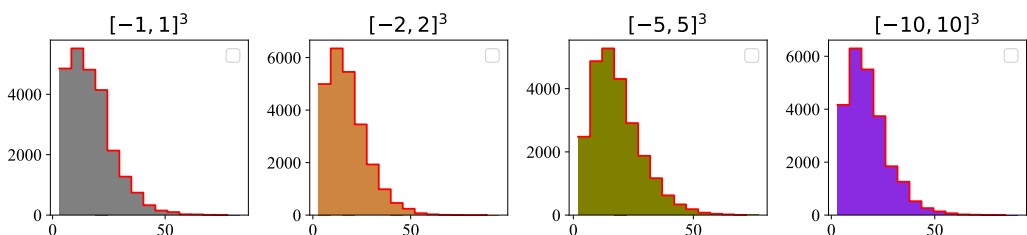

Figure 15: The simplicity and uniformity hold true for larger bounding boxes under the Xavier uniform initialization.

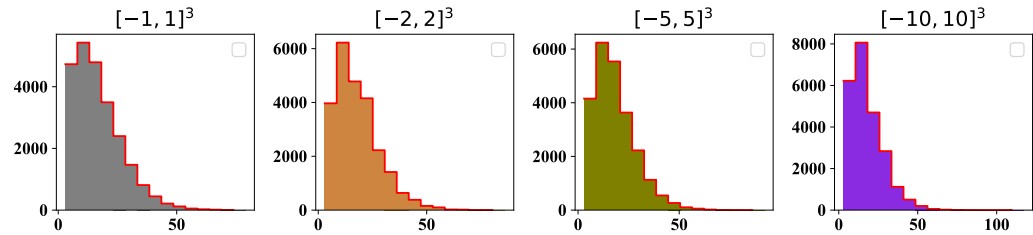

Figure 16: The simplicity and uniformity hold true for larger bounding boxes under the Xavier normal initialization.

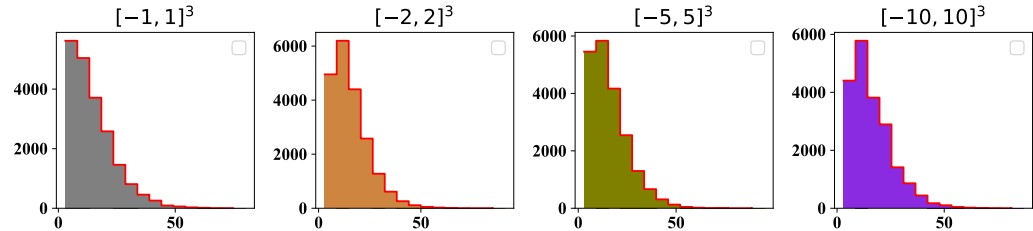

Figure 17: The simplicity and uniformity hold true for larger bounding boxes under Kaiming initialization.

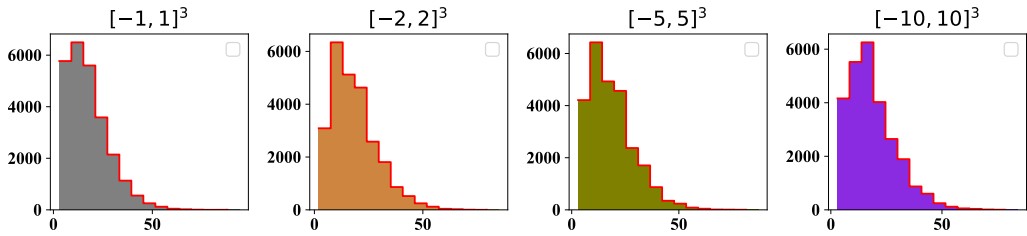

Figure 18: The simplicity and uniformity hold true for larger bounding boxes under the orthogonal initialization.

In the above experiments, we set the outer bounding box to $[-1, 1]^d$, where $d$ is the dimension. *Will the size of the bounding box change the uniformity of linear regions?* Potentially, a larger bounding box will include more regions, and these regions may be complicated. To resolve this ambiguity, we derive the linear regions of the network 3-80-40-1 when setting the bounding box size to $[-1, 1]^3, [-2, 2]^3, [-5, 5]^3, [-10, 10]^3$, respectively. Earlier, we have shown that different bias values do not affect the uniformity and simplicity of polytopes. Therefore, here we randomly set the bias value to 0.01. The initialization methods are the Xavier uniform, Xavier normal, Kaiming, and orthogonal initialization. The results are plotted in Figures 15, 16, 17, and 18, from which we have two observations. First, for different initialization methods, when the size of the outer bounding box

increases, the number of polytopes increases. This is probably because more polytopes are included in the larger area. Second, we find that the uniformity and simplicity of polytopes hold for both smaller and larger bounding boxes for different initialization methods.

# G SUPPLEMENTARY EXPERIMENTS FOR NETWORK STRUCTURES

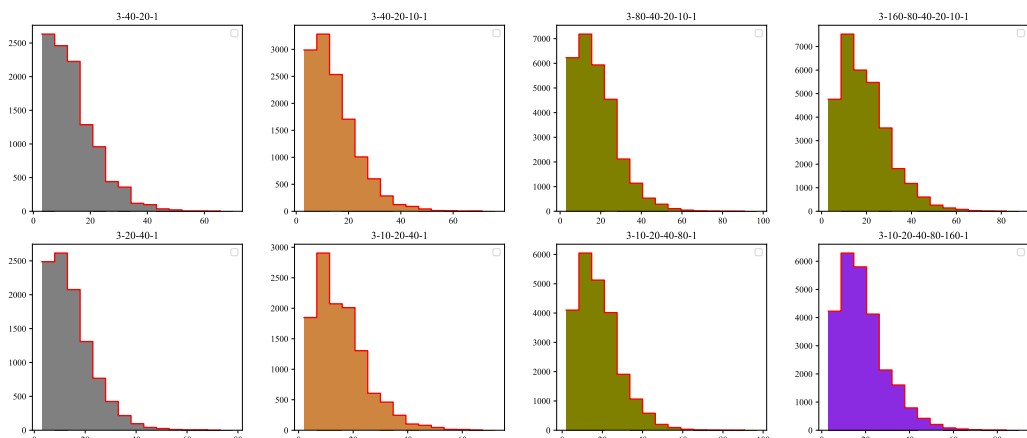

Figure 19: The simplicity and uniformity hold true for both pyramidal and inverted pyramidal structures under the Xavier uniform initialization.

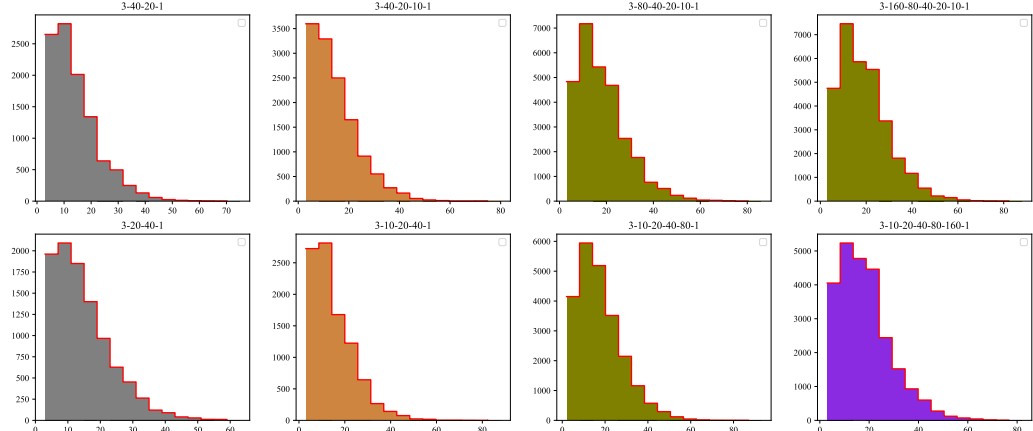

Figure 20: The simplicity and uniformity hold true for pyramidal and inverted pyramidal structures under the Xavier normal initialization.

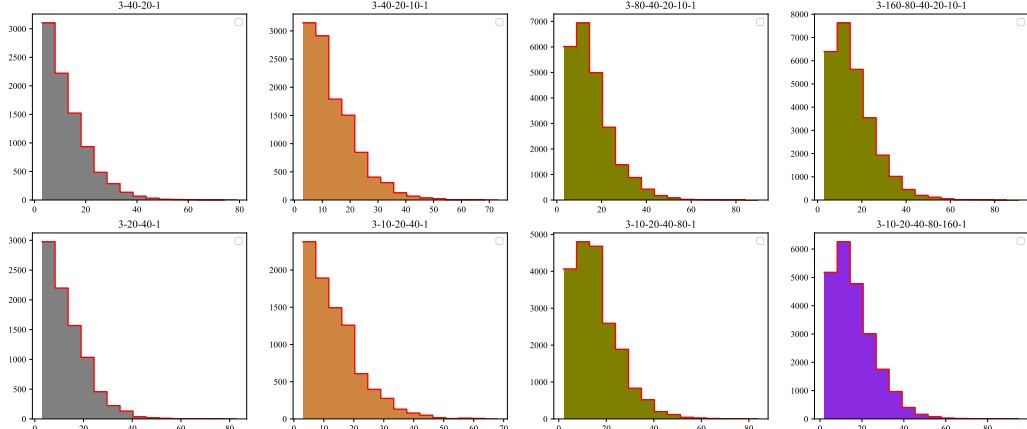

Figure 21: The simplicity and uniformity hold true for pyramidal and inverted pyramidal structures under the Kaiming initialization.

In the above experiments, the structures of all networks we use are pyramidal. Here, we investigate how pyramidal and inverted pyramidal structures affect the uniformity and simplicity of polytopes.

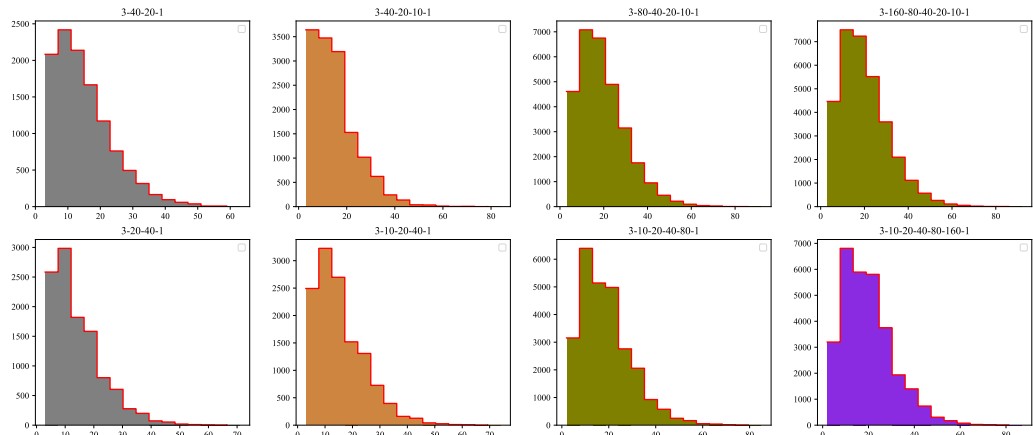

Figure 22: The simplicity and uniformity hold true for pyramidal and inverted pyramidal structures under the orthogonal initialization.

Earlier, we have shown that different bias values and bounding boxes do not undermine the uniformity and simplicity of polytopes. Therefore, here we randomly set the bias value to 0.01, and the bounding box size to $[-1, 1]^3$. A total of 8,000 points are uniformly sampled from $[-1, 1]^3$ to compute the polytope. At the same time, we check the activation states of all neurons to avoid counting some polytopes more than once. The initialization methods are the Xavier uniform, Xavier normal, Kaiming, and orthogonal initialization. The compared network structures are (3-40-20-1, 3-20-40-1), (3-40-20-10-1, 3-10-20-40-1), (3-80-40-20-1, 3-20-40-80-1), and (3-160-80-40-20-10-1, 3-10-20-40-80-160-1). The histograms are shown in Figures 19, 20, 21, and 22. First, we find that given the same number of neurons, for different initialization methods, the total number of polytopes generated by inverted pyramidal strcutures is smaller than that of pyramidal structures. This might be because a neural network is a sequential model, the earlier layer forms a basis for layer layers to cut. The earlier layers with more neurons can have more hyperplanes, which can facilitate more polytopes. Second, for different initialization methods, the uniformity and simplicity of polytopes are respected by both pyramidal and inverted pyramidal structures.

# H SUPPLEMENTARY EXPERIMENTS FOR BOTTLENECKS

Here, we investigate if the bottleneck layer in a network will affect the uniformity and simplicity of polytopes. We randomly set the bounding box size to $[-1, 1]^3$, and the bias value to 0.01. A total of 8,000 points are uniformly sampled from $[-1, 1]^3$ to compute the polytope. At the same time, we check the activation states of all neurons to avoid repetitive calculation. The initialization methods are the Xavier uniform, Xavier normal, Kaiming, and orthogonal initialization. The network structures with bottlenecks are 3-20-10-20-1, 3-20-10-10-20-1, 3-20-20-10-20-20-1, and 3-20-20-10-10-20-20-1. The histograms are shown in Figures 23, 24, 25, and 26. We find that for different initialization methods, the polytopes generated by a network with bottleneck layers are still uniform and simple.

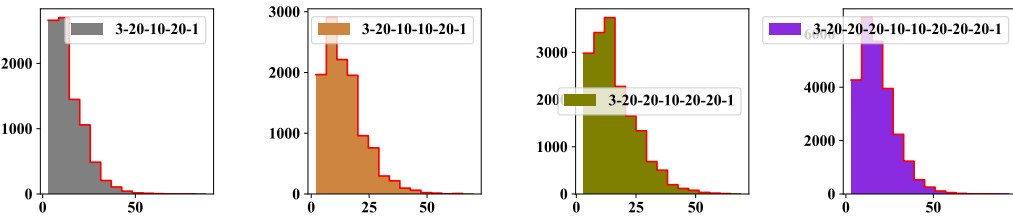

Figure 23: The simplicity and uniformity hold true for bottlenecks under Xavier initialization.

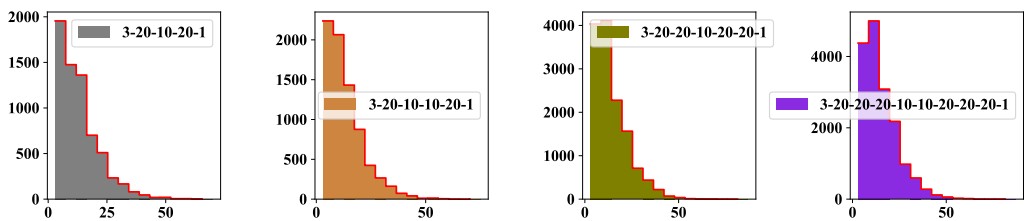

Figure 24: The simplicity and uniformity hold true for bottlenecks under the Xavier normal initialization.

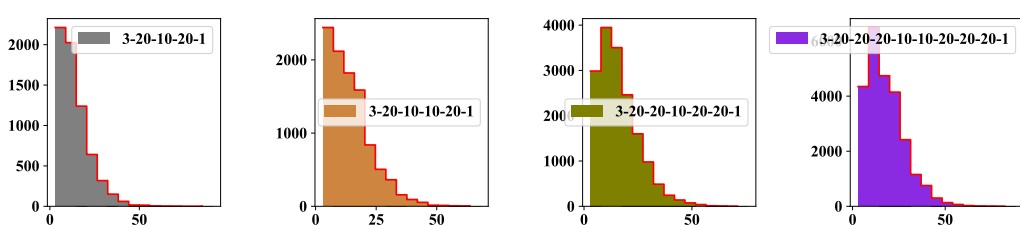

Figure 25: The simplicity and uniformity hold true for bottlenecks under the Kaiming initialization.

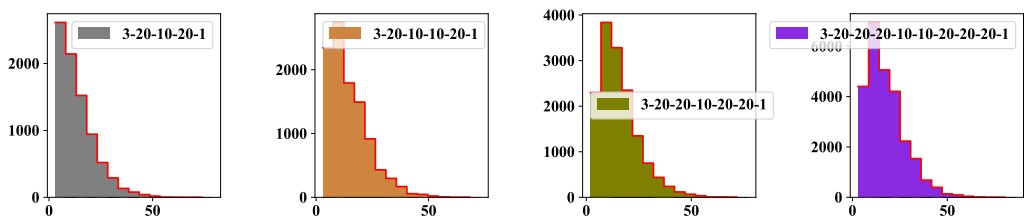

Figure 26: The simplicity and uniformity hold true for bottlenecks under the orthogonal initialization.

# I  SUPPLEMENTARY EXPERIMENTS FOR DIFFERENT DIMENSIONS

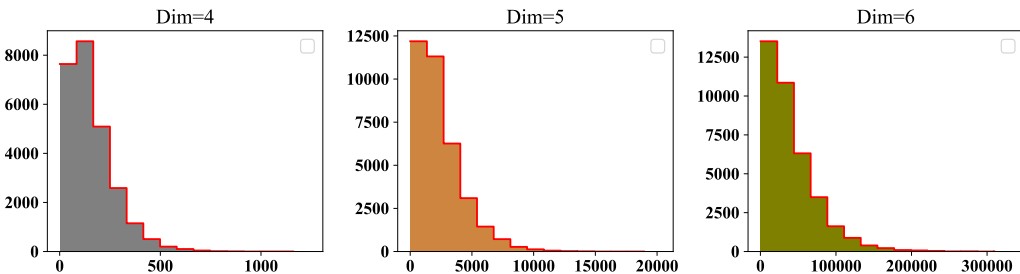

Figure 27: The simplicity and uniformity hold true for different dimensions under Xavier initialization

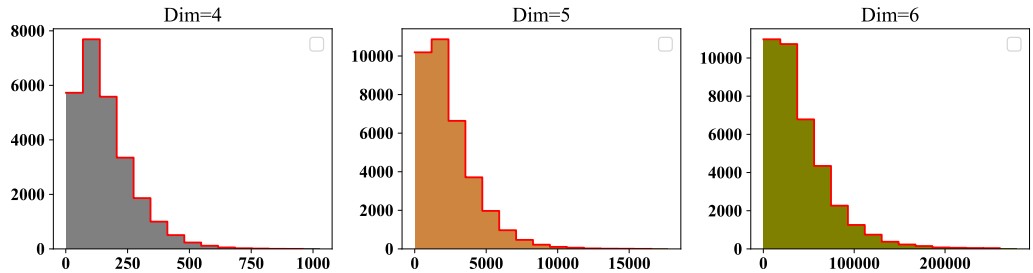

Figure 28: The simplicity and uniformity hold true for different dimensions under Xavier normal initialization

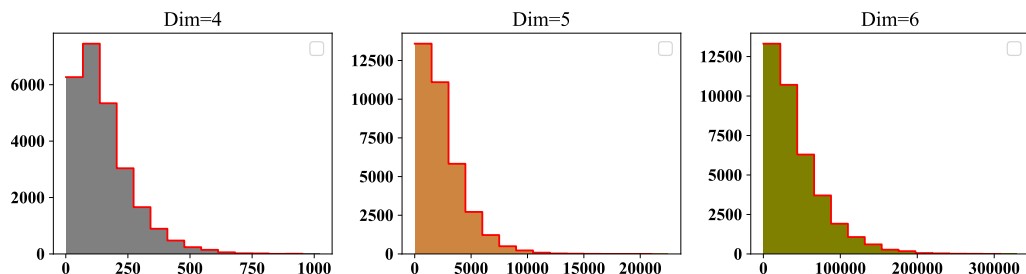

Figure 29: The simplicity and uniformity hold true for different dimensions under Kaiming initialization

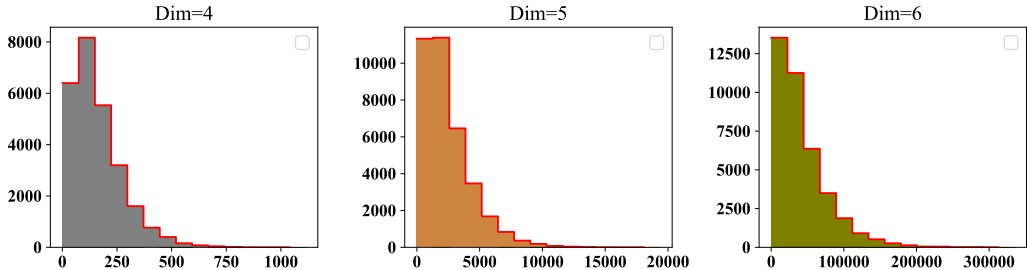

Figure 30: The simplicity and uniformity hold true for different dimensions under orthogonal initialization

Here, we investigate how the input dimension will affect the distribution of polytopes. We compute the polytopes of three networks: 4-40-20-1, 5-40-20-1, and 6-40-20-1. Due to the intrinsic difficulty of computing simplices in high dimensional space, it is very time-consuming to go higher dimensions. We set the bias values to $0.01$. The outer bounding box is $[-1, 1]^d$, where $d$ is the dimensionality. A total of 8,000 points are randomly sampled from $[-1, 1]^d$. At the same time, we check the activation states of all neurons to avoid computing some polytope more than once. The initialization methods are the Xavier uniform, Xavier normal, Kaiming, and orthogonal initialization. Figures 27, 28, 29, and 30 show the distribution of #simplices. As the dimension increases, a polytope tends to have much more simplices, while the total number of polytopes only slightly

increases. We use the triangulation method to compute the number of simplices. According to triangulation properties, the maximum number of simplices approximately equals $\mathcal{O}(M^{d/2})$, where $M$ is the number of simplices, and $d$ is the dimension. This is why the complexity of the linear regions will increase. For different initialization methods, most polytopes are those with a smaller number of simplices.

