# OpenReview forum: "Beyond Counting Linear Regions of Neural Networks, Simple Linear Regions Dominate!"
_ICLR.cc/2023/Conference — Submitted to ICLR 2023_

### Official Review · Reviewer_KuXu · 2022-10-24

**Confidence:** 4
**Correctness:** 4
**Technical Novelty And Significance:** 2
**Empirical Novelty And Significance:** 2
**Recommendation:** 3

**Clarity, Quality, Novelty And Reproducibility:**

The paper is clearly written.

The originality is questionable. In fact, the paper adds one more observation to the earlier work by Hanin & Rolnick (2019b): Hanin & Rolnick (2019b) showed that deep ReLU networks have few polytopes, and this paper shows that such polytopes have to be simple (namely with small number of simplices). The theoretical results are interesting, although they remain at the level of a (cute) observation. The numerical results only deal with networks at initialization.

**Strength And Weaknesses:**

Strengths

* To the best of my knowledge, the number of simplices has not been investigated before, as a measure of the inherent simplicity of the solution found by a ReLU network.

* The numerical results show that, at initialization, the polytopes of deep ReLU networks consistently have triangulations composed by few simplices.

Key weaknesses

* The experimental results only concern initialization, and there is a single qualitative experiment concerning the training of the neural network (Figure 8). It is hard to draw any conclusions on the training dynamics in absence of more quantitative results about gradient descent training.

* The theoretical results are interesting, but relatively straightforward and, given the lack of a more formal connection with gradient training, the insight they bring on the phenomenon of implicit bias remains questionable.



Minor weaknesses


* Because of the inherent difficulty of the counting problem, only relatively small input dimensions can be investigated. This makes it very difficult to scale the experiments of this work to any setting where the input dimension is closer to practice.

* The authors seem to always consider the regime in which the number of neurons $n$ is large and the input dimension $d$ is constant. What happens when $d$ is allowed to scale with $n$? In this case, it seems that both the upper and the lower bound of Theorem 1-2 are dominated by $\mathcal O(n^{dL-1})$. Is it be possible to understand the constant hidden in the $\mathcal O(\cdot)$? Do the results of Theorem 3-4 still hold if $d$ is allowed to grow with $n$?

* Can the results be generalized to deep networks in which the number of neurons in the various layers differ? Is some interesting behavior happening in the presence of a bottleneck (i.e., a layer with few neurons in the middle of two layers with many more neurons)?


**Summary Of The Paper:**

This paper studies the number of simplices contained in triangulations of all polytopes generated by ReLU networks. From the theoretical viewpoint, the authors prove upper (Theorem 1) and lower (Theorem 2) bounds on the maximal number of simplices. This implies that the average number of faces in a polytope grows only linearly in the input dimension (Theorem 3 and 4, for neural networks with one and multiple hidden layers, respectively). From the experimental viewpoint, the authors demonstrate that, at initialization, most of the polytopes are simple (i.e., have few faces). The authors vary the initialization method, the depth, the dynamic range of the input, the size of the biases and (to the extent allowed by numerical simulations) the input dimensions, showing that their conclusions hold in rather large generality.

**Summary Of The Review:**

The paper provides an observation about the number of simplices in the polytopes of deep ReLU networks. Even if this observation is novel, the connection to gradient descent training and its implicit bias is not sufficiently well discussed to bring the paper above the acceptance bar. Another (minor) weakness is that the results hold only for networks in which the input dimension is significantly smaller than the number of neurons, and this number of neurons is the same in every layer.

---

> ### Author Response · Authors · 2022-11-18
> **Response to Reviewer KuXu**
>
> Dear Reviewer KuXu:
>
> We would like to thank you for your recognition of our discovery and theory. We have addressed all of your concerns in the rebuttal, and also will revise the paper following your comments. We are looking forward to discussing this draft with you further. If you are satisfied with our response, please consider raising your score.
>
>
> **[Q1] The experimental results only concern initialization. It is hard to draw any conclusions on the training dynamics in absence of more quantitative results about gradient descent training.**
>
> Thanks for this suggestion. We validate if the uniformity and simplicity of linear regions will be broken during training. We answer this question by training a fully-connected network using ReLU activation function on a real-world problem and counting the simplices of polytopes. The task is to predict if a COVID-19 patient will be at high risk, given one's health status, living habits, and medical history. The results (Section 4.2 of the rebuttal revision) show that throughout the training, most polytopes are simple.
>
>
>
> **[Q2] The theoretical results are interesting but relatively straightforward and, given the lack of a more formal connection with gradient training, the insight they bring on the phenomenon of implicit bias remains questionable.**
>
> In our proof, we only assume that hyperplanes are in general position and the number of neurons is very large in a layer for a multi-layer network. Therefore, these theorems are the general description of what kind of functions a network tends to learn, regardless that a network is trained or just randomly initialized.
>
>
> **[Q3] Only relatively small input dimensions can be investigated.**
>
> We agree with you that it is computationally prohibitive to empirically verify our conclusion in high-dimensional space. However, our theorems imply that our conclusion still holds true, as we don't have any assumptions on dimension (d). The average number of hyperplanes is smaller than $2d+1$.
>
>
> **[Q4] The authors seem to always consider the regime in which the number of neurons is large and the input dimension  is constant. What happens when  is allowed to scale with ? In this case, it seems that both the upper and the lower bound of Theorem 1-2 are dominated by . Is it be possible to understand the constant hidden in the ? Do the results of Theorem 3-4 still hold if  is allowed to grow with ?**
>
> Yes, the upper and the lower bound of Theorem 1-2 are dominated by $\frac{2n^{dL}}{(d-1)!(d!)^{L-1}}$ and $\frac{n^{dL}}{d^{d(L-1)}d!}$ respectively. Furthermore, the coefficients of lower terms can be derived by Section B.3 in the supplemental material. Thanks for pointing out the case when $d$ is large enough, which is what we would like to explore in the future.
>
> **[Q5] Can the results be generalized to deep networks in which the number of neurons in the various layers differ? Is some interesting behavior happening in the presence of a bottleneck**
>
> Yes, the results can be generalized to deep networks in which the number of neurons in the various layers differ. In this case, we still need to assume that the number $n_i$ in the $i$-th layer is large than input dimension $d$. By this we can derive that the upper and the lower bound of Theorem 1-2 are dominated by $\frac{2\left(\prod_{i=1}^L n_i\right)^{d}}{(d-1)!(d!)^{L-1}}$ and $\frac{\left(\prod_{i=1}^L n_i\right)^{d}}{d^{d(L-1)}d!}$ respectively. We also empirically verify our conclusion in Appendix H.
>
>
> **[Q5] The originality is questionable. In fact, the paper adds one more observation to the earlier work by Hanin \& Rolnick (2019b)**
>
> We agree that both our work and Hanin \& Rolnick (2019b) focus on the polytopes produced by a ReLU network. However, our work is fundamentally different from Hanin \& Rolnick (2019b) in the following aspects:
>
> - (Beyond counting) Figuring out the properties of polytopes of a neural network is of fundamental importance for the understanding of neural networks, however, Hanin \& Rolnick (2019b) only stays at the level of counting the number of polytopes, which blocks us from gaining other valuable findings. Our finding specifically reveals the shape information of a polytope, which is a big stride forward.
>
> - (Revealing local information) Hanin \& Rolnick (2019b)'s finding is global information regarding a network, while our finding goes from local to global. Given a sample, our results shed light on what kind of polytope this sample lies in, i.e., is it a simple polytope or a complicated polytope? Such local information is useful to analyze the interpretability and robustness of a network.

---

> > ### Comment · Reviewer_KuXu · 2022-11-26
> > **follow up**
> >
> > I would like to thank the authors for the thoughtful responses and for editing the paper. However, my main concern regarding the conclusions to be drawn about gradient descent remains essentially unaddressed, and the rather simple experiment proposed by the authors is not sufficiently convincing in that regard. Q4 (i.e., what happens when $d$ is allowed to scale with $n$) remains unaddressed as well.
> >
> > Finally, I find Weakness 1 pointed out by reviewer rZ3w to be rather important: it is not clear why the number of simplices is a proper measure of complexity (and, hence, why e.g. it would lead to an interesting implicit bias for gradient descent).
> >
> > For these reasons, I have decided to keep my score.

---

### Official Review · Reviewer_H4R5 · 2022-10-25

**Confidence:** 3
**Clarity, Quality, Novelty And Reproducibility:** 1. Clarity
**Correctness:** 3
**Technical Novelty And Significance:** 2
**Empirical Novelty And Significance:** 2
**Recommendation:** 5

**Details Of Ethics Concerns:**

There are no Ethics Concerns.

**Strength And Weaknesses:**

Strength:

1. The research problem of the paper is very meaningful for deep learning.

2. The paper explains the relationship between ReLU and deep neural network overfitting.

3. The paper gives proof of the theorem. This makes sense for theoretical explanations of deep neural networks.

Weaknesses:

1. The paper explains why deep learning does not overfit. This is contrary to my previous perception. I want to ask what assumptions did the author make when proving this conclusion? Are these assumptions adequately accounted for in the paper?

2. The paper assumes that the input space of an NN is a d-dimensional hypercube. What is the correlation between this assumption and the
convolutional neural networks? If The author's analysis here seems only to contain fully connected layers?

3. What implications do the authors' conclusions have for subsequent research? I'm a little confused about what kind of inspiration this proof can bring us.

4. The authors show that deep neural networks do not overfit. However, this conflicts with existing cognition. Has the author analyzed this conflict? Under what circumstances will deep neural networks not overfit?

5. What is the meaning of 3-4-1 in Table 1? I'm a little confused.

**Summary Of The Paper:**

1) The paper establishes a theorem to explain why polyhedra produced by deep networks are simple and uniform and the linear region dominates. 2) The paper proves that deep learning does not overfit. 3) The paper promotes the theoretical research of ReLU. 4) The paper is submitted with a basic implementation.

**Summary Of The Review:**

I am not very clear about some details and derivations of the paper. In addition, I do not know the impact of the paper on subsequent research work. I would like the author to reply to my question. Also, I would like to discuss this with the other reviewers to determine the final score.

---

> ### Author Response · Authors · 2022-11-18
> **Response to Reviewer H4R5**
>
> Dear Reviewer H4R5:
>
> We would like to thank you for your recognition. We are sorry for some misunderstandings in the current version of the paper. We have addressed all of your concerns in the rebuttal, and also will revise the paper following your comments. We are looking forward to discussing this draft with you further.
>
>
> **[Q1] The paper explains why deep learning does not overfit. This is contrary to my previous perception. I want to ask what assumptions did the author make when proving this conclusion? Are these assumptions adequately accounted for in the paper?**
>
> A network used in practice is highly over-parameterized compared to the number of training samples. Based on the conventional generalization theory, such a highly over-parameterized network will incur severe overfitting. But in practice, deep learning performs rather well. To solve this puzzle, extensive studies have proposed that a network is implicitly regularized to learn a simple (not more complicated than necessary) solution. Implicit regularization is also referred to as an implicit bias. Regardless, we do not say that a deep network is completely free of over-fitting in any scenario. Our result reveals what kind of functions a network tends to learn, which exerts no assumptions.
>
> **[Q2] The paper assumes that the input space of an NN is a d-dimensional hypercube. What is the correlation between this assumption and the convolutional neural networks? If The author's analysis here seems only to contain fully connected layers?**
>
> Thanks for this suggestion. This assumption is added to the input. Therefore, it can be extended to the analysis of convolutional neural networks.
>
>
> **[Q3] What implications do the authors' conclusions have for subsequent research? I'm a little confused about what kind of inspiration this proof can bring us.**
>
> ReLU is the most popular activation function in deep networks. It has been analyzed that a ReLU network is a piecewise linear function, and it partitions the space into convex polytopes. Our result suggests that polytopes formed by a ReLU network are encompassed by surprisingly few hyperplanes, which is helpful to develop algorithms to compute hyperplanes of polytopes, thereby mathematically exactly grasping what kind of functions a network learns. Then, almost all interpretations of a deep network can be catalyzed.
>
>
> **[Q4] What is the meaning of 3-4-1 in Table 1? I'm a little confused.**
>
> Sorry for the confusion. They are network structures. For a network structure $X-Y_1$-$\cdots$-$Y_H-1$, $X$ represents the dimension of the input, and $Y_h$ is the number of hidden neurons in the $h$-th hidden layer.

---

### Official Review · Reviewer_rZ3w · 2022-10-30

**Confidence:** 4
**Correctness:** 3
**Technical Novelty And Significance:** 2
**Empirical Novelty And Significance:** 1
**Recommendation:** 3

**Clarity, Quality, Novelty And Reproducibility:**

Clarity:

The paper is mostly clear on its technical contributions, but the discussions relating the results to actual phenomena in neural nets like generalisation and spectral bias is mostly imprecise.



**Strength And Weaknesses:**

Strength:

1. A good mix of theoretical and empirical results for counting the number of simplices.

2. Simplifying the core argument of "expected number of polytopes" in Hanin et al. to a simple general position argument clears up some mental space for other arguments.

Weaknesses:

1. The argument that the number of simplices or faces of the linear regions a measure of complexity seems reasonable but needs more validation. e.g. there are easy VC type bounds linking the number of pieces in a piecewise linear function family to generalisation bounds, similar bounds for the number of simplices (without going through the number of linear regions) would be valuable.

2. The statement for Theorem 2 is slightly confusing. "Maximal number of simplices is at least ..." what is the maximum over? Do dring the dependence of the different variables clearly.

3. The upper and lower bounds are quite tight (assuming general position assumptions), but the number seems too large to make any reasonable conclusion for even some moderately sized networks -- conditions on the parameters of the neural under which this can be significantly smaller (or larger) would give some insight into complexity of neural nets.

4. The only constraint on the parameters I can see is its requirement to be in general position. For one layer nets that happens with probability 1 under typical inits. But what about the condition in Theorem 4 that generalises to multiple hidden layers? Can you argue that this also happens with probability 1 under typical initialisations?

5. A study on when (for what parameter configurations especially) is the number of simplices small/large would be more valuable, when juxtaposed with other important properties of neural nets, such as generalisation/frequency bias. e.g. questions like are there more simplices after training with random data than true labels? e.g a ranking study of different networks of the same architecture (and same init) trained on same data (but different mini-batches or different learning rates or different optimisation algorithms) on the basis of generalisation error and number of simplices. Do the two rankings agree?


**Summary Of The Paper:**

It is well known that the function represented by a ReLU network is piecewise linear, with each linear region corresponding to an activation pattern. There have been papers showing that despite the potential number of possible activation patterns being exponential in the number of neurons, the actual number for a typical randomly initialised network is only polynomial in the number of neurons. This paper extends the idea further and argues that the number of faces of the polyhedral linear regions is also small, and hence typical neural nets are not as complicated as the worst case.

**Summary Of The Review:**

The paper attacks a novel question, whose significance is of questionable value.

---

> ### Author Response · Authors · 2022-11-18
> **Response to Reviewer rZ3w**
>
> Dear Reviewer rZ3w:
>
> We would like to thank you for your recognition. We are sorry for some misunderstandings in the current version of the paper. We have addressed all of your concerns in the rebuttal, and also will revise the paper following your comments. We are looking forward to discussing this draft with you further.
>
>
>
> **[Q1] The argument that the number of simplices or faces of the linear regions a measure of complexity seems reasonable but needs more validation. e.g. there are easy VC type bounds linking the number of pieces in a piecewise linear function family to generalisation bounds, similar bounds for the number of simplices (without going through the number of linear regions) would be valuable.**
>
> Thanks for the question. Yes, for VC dimesion, there are many results exploring its relation with generalisation bounds. For the number of simplices, we also would like to discover its relation with generalisation bounds in future works.
>
>
> **[Q2] The statement for Theorem 2 is slightly confusing. "Maximal number of simplices is at least ..." what is the maximum over? Do dring the dependence of the different variables clearly.**
>
> For each given group of parameters of a DNN, we can calculate a number of simplices. When the parameters change, the number of simplices also changes. Thus the maximal number of simplices means the largest number of simplices we can obtain when the parameters change.
>
> **[Q3] The upper and lower bounds are quite tight (assuming general position assumptions), but the number seems too large to make any reasonable conclusion for even some moderately sized networks -- conditions on the parameters of the neural under which this can be significantly smaller (or larger) would give some insight into complexity of neural nets.**
>
>
> When the network has only one hidden layer, the upper and lower bounds can be verified in our experiments. For multi-layer NNs, we admit that usually the upper and lower bounds are very large thus difficult to verify in experiments. Deriving upper and lower bounds after adding some conditions on the parameters is what we would like to consider in the future.
>
> **[Q4] The only constraint on the parameters I can see is its requirement to be in general position. For one layer nets that happens with probability 1 under typical inits. But what about the condition in Theorem 4 that generalises to multiple hidden layers? Can you argue that this also happens with probability 1 under typical initialisations?**
>
> Yes, intuitively, for multiple hidden layers, we can still obtain the hyperplanes in general positions with probability 1, since it just corresponds to adding random hyperplanes in each polytope (although we admit that it can not be easily proved).

---

### Official Review · Reviewer_htwS · 2022-11-05

**Confidence:** 4
**Correctness:** 3
**Technical Novelty And Significance:** 3
**Empirical Novelty And Significance:** 3
**Recommendation:** 3

**Clarity, Quality, Novelty And Reproducibility:**

Here are some detailed points about the writing:
- What the authors describe as a face starting in the bold part at the end Preliminary 1 could more precisely be called a *facet*.
- The description of a polytope in Preliminary 3 is a little confusing with so many commas for different purposes in the set.
- In Theorem 2, I would advise the authors against using the term *maximal* if *maximum* would be more precise (they don't mean the same thing).
- Most plots presented in the paper lack axis titles or at least a description in the caption for what the axis title of each plot should be.
- I believe the use of "surprisingly" in the title of Section 4 is completely unnecessary and actually cliché for imitating previous papers.
- In the second line of Section 4, I believe you meant to say "linear regions" when you wrote "simplices".
- The use of "thing" and "pretty small" in Remark 1 is too informal and uninformative.

**Strength And Weaknesses:**

Strengths:
- The introduction is very well written and gives great context for the problem of analyzing the complexity of neural networks under the optics of linear regions.
- The idea of studying the complexity of linear regions in terms of their simplicial decomposition seem novel and relevant.
- If confirmed, the results claimed by the authors provide further insight into what functions we can expect to obtain from neural networks.

Weaknesses:
- The numerical experiments lack consistency. For example, forms of initialization change between one experiment and another without any justification (sometimes 4 are used, sometimes only Xavier, sometimes only Kaiming).
- Although Section 4 has ReLU in its title, Subsection 4.2 discusses results with maxout (why?).
- The only experiment in which the input dimension goes beyond 3 clearly show some gain in complexity of the linear regions, but this seems to be ignored by the authors.
- As far as I can understand, each plot is based on a single network, which is far from ideal.
- It is not clear to me what the authors mean by the linear regions being uniform in Section 4.
- I am curious and puzzled about Theorem 4. Given that linear regions are typically in general position in shallow networks, which implies a large number of linear regions and consequently simpler linear regions, I can easily buy Theorem 3. However, the number of linear regions drops when the network gets deeper. I don't see any intuition for why Theorem 4 should be true.


**Summary Of The Paper:**

This paper refines the study of linear regions in neural networks by analyzing how each linear region can be decomposed into simplices. The authors argue through theoretical results and experiments that most linear regions can be decomposed into very few simplices; hence implying in yet another form that neural networks end up producing simple functions despite their potential to also represent much more complex functions.

**Summary Of The Review:**

The paper brings interesting ideas and communicates them in an accessible way for the most part. However, it is not clear to me if all results are correct and whether their experiments reflect what would happen in networks with larger inputs.

---

> ### Author Response · Authors · 2022-11-18
> **Response to Reviewer htwS**
>
> Dear Reviewer htwS:
>
> We would like to thank you for your recognition and comments. We are sorry for some misunderstandings in the current version of the paper. We have addressed all of your concerns in the rebuttal, and also will revise the paper following your comments. For more details, please check out Appendices D-I in the rebuttal revision. We are looking forward to discussing this draft with you further. If you are satisfied with our response, please consider raising your score.
>
>
> **[Q1] The numerical experiments lack consistency. For example, forms of initialization change between one experiment and another without any justification (sometimes 4 are used, sometimes only Xavier, sometimes only Kaiming).**
>
> Because our experiment results (Figure 3) show that different initialization methods do not affect the uniformity and simplicity of polytopes, in the later experiments, we randomly choose the initialization method. In this rebuttal revision, per your question, we have redone the experiments under four different initialization methods for consistency. Please check out our new results in Appendices D-I which we conclude that for different network depths, sizes of the outer bounding box, biases, the bottleneck, network architecture, and input dimensions, simple polytopes still dominate among all polytopes of a network.
>
>
> **[Q2] Although Section 4 has ReLU in its title, Subsection 4.2 discusses results with maxout (why?).**
>
> Sorry for the confusion. We have done the same experiment by using ReLU activation and updated the associated figure. With ReLU activation, the observation for the shapes of polytopes is consistent. Almost all the polytopes are triangles or quadrilaterals. Although these polytopes are from a cross-section other than the whole landscape, one can indirectly sense the simplicity of these polytopes.
>
> **[Q3] The only experiment in which the input dimension goes beyond 3 clearly show some gain in complexity of the linear regions, but this seems to be ignored by the authors.**
>
> After the vertex enumeration method is applied to derive the vertices of a polytope, we use the triangulation method to compute the number of simplices. According to the properties of triangulation, the maximum number of simplices approximately equals $M^{d/2}$, where $M$ is the number of simplices, and $d$ is the dimension. This is why the complexity of the linear regions will increase. We have added this discussion in this rebuttal revision.
>
> **[Q4] It is not clear to me what the authors mean by the linear regions being uniform in Section 4.**
>
> Here, the uniformity and simplicity mean that although theoretically quite diverse and complicated polytopes can be derived, simple polytopes dominate.
>
>
> **[Q5] However, the number of linear regions drops when the network gets deeper. I don't see any intuition for why Theorem 4 should be true.**
>
>
> Thanks for pointing out this question. Yes, Theorem 3 is always true if the hyperplanes are in general position in shallow networks. For deep networks in Theorem 4, the number of linear regions increases when the network gets deeper.
> To prove that the average number of faces in linear regions is still small, the only assumption we make is that the number of hyperplanes in each region $S$ generated by hidden neurons in the $i$-th layer is large enough, and these hyperplanes are always in general position. This assumption is often used in deep learning theories such as neural network Gaussian process and neural tangent kernel. Based on our experiments, we believe Theorem 4 is still true without such an assumption, this is what we would like to prove in future works.
>
>
> **[Q6] Clarity issues about the writing**
>
> Per your comments, all informal/imprecise expressions were edited.
>
>
> - .... could more precisely be called a facet. (Edited).
>
> - The description of a polytope in Preliminary 3 is a little confusing with so many commas for different purposes in the set. (Edited).
>
> - using the term maximal if maximum would be more precise (they don't mean the same thing). (Corrected them all).
>
> - Most plots presented in the paper lack axis titles or at least a description in the caption for what the axis title of each plot should be. (We have clarified this at the end of the first paragraph in Section 4.1. "Hereafter, if no special specification, the x-axis of all figures denotes the number of simplices a polytope has, and the y-axis denotes the count of polytopes with a certain number of simplices.")
>
> - I believe the use of "surprisingly" in the title of Section 4 is completely unnecessary and actually cliché for imitating previous papers. (Deleted).
>
> - In the second line of Section 4, I believe you meant to say "linear regions" when you wrote "simplices". The use of "thing" and "pretty small" in Remark 1 is too informal and uninformative. (edited "thing" with "property"; edited "pretty" with "rather").
>
> Sincerely
>
> Anonymous authors

---

> > ### Comment · Reviewer_htwS · 2022-11-22
> > **Following up**
> >
> > I appreciate the effort of the authors in improving the paper. I believe that I was not entirely clear in the point that you addressed as Q5. The number of linear regions does increase if you add another layer and everything else is the same. However, if you compare the number of linear regions when a fixed number of neurons is put in a single layer (say a shallow network with width 100) with a network having multiple layers with smaller width each (say a deep network with 10 layers and width 10, hence also 100 neurons), the number of linear regions in deeper networks tend to be smaller. See, for example, Figure 5 in https://arxiv.org/pdf/1711.02114.pdf .
> >
> > The fact that the hyperplanes are in general position in subsequent layers does not make up for the fact that the input of the subsequent layers is often confined to subspaces of much smaller dimension, and the hyperplanes are unlikely to be in general position within those subspaces.
> >
> > Perhaps it might be true that the shape of the linear regions remain simple in deeper networks, but I believe you would need to work with weaker assumptions to prove that. One alternative would be to invest in other experimental approaches to see how their shape looks like, so that you explore this for inputs that are reasonably larger. That brings me back to what you addressed as Q3: as soon as the size of the input is a little larger, we see a lot more complexity going on than the message in your paper seems to suggest.

---

### Decision · Program_Chairs · 2023-01-20

**Decision:**

Reject

**Justification For Why Not Higher Score:**

The reviewers give good reasons why they did not give this article a higher score.

I would like to add a few observations that I hope the authors might find useful.

* Contrary to a claim on pg 2, it is not clear how a direct application of Zaslavsky's theorem could be used to obtain a non-trivial lower bound on the maximum number of linear regions of a deep ReLU network. The referenced work uses an explicit construction.
* It would be useful to note that for the subdivision of input space the work of Hanin and Rolnick and later also Tseran and Montufar for maxout estimated the number of faces of any dimension, including faces of co-dimension 1.
* Contrary to a claim on pg 1 and pg 2, the subdivision of input space into regions does not necessarily consist of polytopes strictly speaking but rather polyhedra which may be unbounded and may have fewer than $n_0+1$ facets.
* On pg 5 it is stated that a "(d-1)-dim face of a d-dim polytope, it can only be a face for one unique simplex in a triangulation of this polytope" and that "thus the total number of simplices ... must be smaller than or equal to the total number of (d-1)-dim faces". However, if a face is not a simplex, then in a triangulation it must be subdivided into simplices, meaning that it will appear on several simplices. Note that for instance a triangulation of a d-cube will need to contain all 2^d vertices of the cube and since each simplex has only d+1 vertices, there will be at least 2^d/(d+1) simplices in a triangulation of the cube, which certainly is larger than the number of (d-1)-dim faces, which is just 2d.
* Triangulations of polytopes are a non trivial subject that has been explored in the literature, which in my view is not adequately discussed in this submission.

**Justification For Why Not Lower Score:**

Taking concerns about correctness into account, the article could be given a lower score.

**Metareview: Summary, Strengths And Weaknesses:**

The submission investigates triangulations of the linear regions of ReLU networks. The idea has interesting aspects but it is not convincingly executed. The reviewers agree that the work is not good enough.



**Summary Of Ac-Reviewer Meeting:**

NA